



# The world Brewer reference triad – updated performance assessment and new double triad

Xiaoyi Zhao[1], Vitali Fioletov[1], Michael Brohart[1], Volodya Savastiouk[2], Ihab Abboud[1], Akira Ogyu[1], Jonathan Davies[1], Reno Sit[1], Sum Chi Lee[1], Alexander Cede[3,4], Martin Tiefengraber[4,5], Moritz Müller[4,5], Debora Griffin[1], Chris McLinden[1]

[1]Air Quality Research Division, Environment and Climate Change Canada, Toronto, M3H 5T4, Canada.
[2]International Ozone Services Inc., Toronto, Canada.
[3]NASA Goddard Space Flight Center, Greenbelt, MD 20771, USA.
[4]LuftBlick, Innsbruck, Austria.
[5]Department of Atmospheric and Cryospheric Sciences, University of Innsbruck, Innsbruck, Austria.

*Correspondence to*: Xiaoyi Zhao (xiaoyi.zhao@canada.ca)

**Abstract.** The Brewer ozone spectrophotometer (the Brewer) was designed at Environment and Climate Change Canada (ECCC) in the 1970s to make accurate automated total ozone column measurements. Since the 1980s, the Brewer has become a World Meteorological Organization (WMO) Global Atmosphere Watch (GAW) standard ozone monitoring instrument. Now, more than 230 Brewers have been produced. To assure the quality of the Brewer measurements, a calibration chain is maintained, i.e., first, the reference instruments are independently absolutely calibrated, and then the calibration is transferred from the reference instrument to the travelling standard, and subsequently from the travelling standard to field instruments. ECCC has maintained the world Brewer reference instruments since the 1980s to provide transferable calibration to field instruments at monitoring sites. Three single-monochromator (Mark II) type instruments (serial numbers #008, #014, and #015) formed this world Brewer reference triad (BrT), and started their service in Toronto, Canada in 1984. In the 1990s, the Mark III type Brewer (known as the double Brewer) was developed, which has two monochromators to reduce the internal instrumental stray light. The double Brewer world reference triad (BrT-D) was formed in 2013 (serial numbers #145, #187 and #191), co-located with the BrT. The first assessment of the BrT's performance was made in 2005, covering the period between 1984 and 2004 (Fioletov et al., 2005). The current work provides an updated assessment of the BrT's performance (from 1999 to 2019) and the first comprehensive assessment of the BrT-D. The random uncertainties of individual reference instruments are within the WMO/GAW requirement of 1 % (0.49 % and 0.42 % for BrT and BrT-D, respectively). The long-term stability of the reference instruments is also evaluated in terms of uncertainties of the key instrument characteristics: the extraterrestrial calibration constant (ETC) and effective ozone absorption coefficients (both having an effect of less than 2 % on total column ozone). Measurements from a ground-based instrument (Pandora spectrometer) and satellites (eleven datasets,



including the most recent high-resolution satellite, TROPOspheric Monitoring Instrument), and reanalysis model (the second Modern-Era Retrospective analysis for Research and Applications, MERRA-2) are used to further assess the performance of world Brewer reference instruments and to provide a context for the requirements of stratospheric ozone observations during the last two decades.

## 1 Introduction

Ozone ($O_3$) is one of the most well-known and critical atmospheric trace gases (WMO, 2018), with remote sensing monitoring of atmospheric ozone being traced back to 1926 (Dobson, 1968). In the late 1970s to early 1990s, stratospheric ozone has become an important scientific topic and a matter of intense interest after discovery and subsequent studies of the Antarctic ozone hole (Farman et al., 1985; Solomon et al., 1986; Stolarski et al., 1986) and ozone depletion on the global scale (Ramaswamy et al., 1992; Stolarski et al., 1991). To perform long-term, automated, ground-based total column ozone monitoring, the Brewer instrument was proposed by Alan Brewer (Brewer, 1973) and developed with James Kerr, Tom McElroy and David Wardle in the early 1980s at Environment and Climate Change Canada (ECCC) (Kerr, 2010; Kerr et al., 1981). In 1988, the Brewer was designated as the World Meteorological Organization (WMO) Global Atmosphere Watch (GAW) standard instrument for total column ozone measurements. ECCC has maintained the world Brewer reference instruments since the 1980s to provide transferable calibration to field instruments at monitoring sites. In practice, three Mark II type instruments (serial numbers #008, #014, and #015) formed this world Brewer reference triad (BrT), and started their service in Toronto (43.781° N, 79.468° W, 187 m a.s.l.), Canada in 1984 (Fioletov et al., 2005). The long-term performance of these three instruments was previously evaluated using direct sun total column measurements for a 20-year period between 1984 and 2004 (Fioletov et al., 2005). Data analysis from this study shows that the random uncertainties of individual observations are within ±1 % in about 90 % of all measurements.

Internal instrumental stray light affects measurements made with the Mark II type instruments; therefore, corrections are applied to the data when necessary (Bais et al., 1996; Fioletov et al., 2000). To significantly reduce this effect, in 1992, ECCC scientists introduced the Brewer Mark III spectrophotometer that uses the same concept of the Mark II model version, but has a second monochromator (Wardle et al., 1996). In 2013, a second world reference standard, known as the double Brewer reference triad (BrT-D), consisting of three Brewer double spectrophotometers (serial numbers #145, #187 and #191) was co-located with the original triad in Toronto (Zhao et al., 2016). The two triads run in parallel. These two triads serve as a calibration reference for travelling standard instruments that are used for calibration of Brewer spectrophotometers deployed across the world in the GAW Programme run under the auspices of the WMO. There are other Brewer triads formed and operated by the Swiss Federal Office of Meteorology and Climatology (Meteo Swiss; the triad is known as the Arosa triad) and the State Meteorological Agency of Spain (AEMET; the triad is known as the Regional Brewer Calibration Center Europe





(RBCC-E) triad). The Arosa triad (Staehelin et al., 1998; Stübi et al., 2017), formed in 1998, was the second Brewer triad worldwide (composed of two Mark II and one Mark III instruments). To better coordinate the Brewer network at the regional scale (León-Luis et al., 2018; Redondas et al., 2018), the RBCC-E triad was formed in 2003 (composed of three Mark III

instruments). The regional reference instruments are regularly compared to the world reference instruments via a travelling standard.

By 2019, there were more than 230 Brewer instruments deployed worldwide within the WMO GAW global ozone monitoring network. From 1999 to 2019 (the period within which the world Brewer reference instruments' data are evaluated in this work),

the World Ozone and Ultraviolet Radiation Data Centre (WOUDC, woudc.org) received Brewer ozone observations from 123 instruments at 88 stations. As a large global monitoring network, the measurement stability is maintained via strict laboratory calibrations (e.g., ozone absorption coefficients from dispersion test) and field calibration (i.e., deriving the extraterrestrial calibration constant). For example, the effective ozone absorption coefficients ($\Delta\alpha$) are determined for each individual instrument in laboratories. The extraterrestrial calibration constant (ETC) has to be determined in the field by one of the two

means: 1) the independent calibration method, i.e., the Langley plot calibration method or the so-called zero airmass extrapolation technique, or 2) the calibration transfer method (e.g., transfer ETC from well-calibrated reference instruments to field instruments). In practice, each field Brewer instrument receives its ETC constant by comparing ozone values with these of the travelling standard instrument. The travelling standard itself is calibrated against the set of world reference instruments (i.e., world Brewer reference triad). Each individual reference instrument is independently calibrated at the Mauna Loa

Observatory (MLO), Hawaii (19.5° N, 155.6° W, 3400 m asl), every 2-6 years (see Table 1) via the Langley plot calibration method. Thus, it is critical to review and assess the world reference instruments' performance on a regular basis.

 Previously, the assessment for the BrT, carried out by Fioletov et al. (2005), examined its twenty-year long record of direct sun (DS) total ozone measurements (1984-2004). It was found that the BrT's precision over these two decades was better than

±1 % (Fioletov et al., 2005). There is no further published assessment for the world reference instruments after that period, and no formal assessment made for the BrT-D yet. In addition, with the increasing number of satellite observations (e.g., OMI, TROPOMI) and ground-based observations from emerging technologies (e.g., Pandora spectrometer) of total ozone columns, it is important to compare the triad datasets with these measurements.

This paper provides a more recent assessment for the BrT (1999-2019) and reports the first assessment of the BrT-D (2013-2019). It is organized as follows: Section 2 describes the ground-based ozone measurements, satellite ozone measurements, and the model reanalysis ozone data. In Section 3, the standard and the new evaluation schemes are introduced, with a detailed description of a new third-party evaluation model. In Section 4, the world Brewer reference instruments (BrT and BrT-D) data products are evaluated by the standard and new schemes. Lastly, Section 5, discusses the challenges for Brewer instruments





to measure ozone at a level better than 1%, in the context of the comparison between the world reference triads, regional triads and high-resolution satellite data. Conclusions are given in Section 6.

## 2 Datasets

### 2.1 Brewer

There are several model versions of the Brewer instrument. The Mark I prototype instruments were tested and operated since

the 1970s (Kerr et al., 1981). The first production version (Mark II) was introduced in the early 1980s. In the 1990s, the double monochromator (Mark III) was developed to reduce the internal instrumental stray light, which allows high-quality total column ozone measurements in large slant column ozone (e.g., low sun elevation) conditions. There were other versions of Brewers developed in the late 1990s (i.e., Mark IV and V) to extend the measuring wavelengths and to measure other trace gases. Today, only the Mark III version of the Brewer is manufactured. Table 1 summarizes some of the specific similarities

and differences between the single and the double Brewer reference triads. More details about Mark II and III's measurement standard deviations and stray light characteristics are provided in Appendix A.





**Table 1. Specific features of single and double Brewer reference triads**

| | Single Brewer | Double Brewer |
|---|---|---|
| **Model Version** | Mark II | Mark III |
| **Serial No.(s)** | #008, #014 and #015 | #145, #187 and #191 |
| **Start of triad observations** | September 1984 | October 2013 |
| **Optical and spectral characteristics** | Single monochromator: a dispersing monochromator with an 1,800 line/mm holographic diffraction grating. | Double monochromator: a top dispersing monochromator with a 3,600 line/mm holographic grating, and a bottom recombining monochromator that is a mirror image of the dispersing monochromator |
| | Spectra measured by a single monochromator that is affected by the internal instrumental stray light in the UV region (Bais et al., 1996; Fioletov et al., 2000). | Significantly less instrumental stray light than in the single monochromators. Thus, increased accuracy of ozone and UV measurements under certain conditions (Bais et al., 1996; Wardle et al., 1996). |
| **Output** | Solar radiation at six UV wavelengths is measured with the spectrometer. The wavelengths are 303.2 nm (almost exclusively for wavelength calibration, i.e., spectral reference test) and five operating wavelengths (306.3 nm, 310.1 nm, 313.5 nm, 316.8 nm and 320.1 nm) used to measure total column ozone and sulphur dioxide using the sun, sky or near full moon as a light source. | |
| | Provides high-quality ozone measurements with a slant ozone column amount up to 1000 DU, which for the global average total ozone column of 300 DU corresponds to an ozone air mass factor of 3.33 and a solar zenith angle (SZA) of about 73° (Zanjani et al., 2019). | Provides high-quality ozone measurements with a slant ozone column amount up to 2000 DU, which for the global average total ozone column of 300 DU corresponds to an ozone air mass factor of 6.67 and a SZA of about 81° (Savastiouk, 2006). |

In general, the Brewer spectrophotometer can provide data products that include column ozone, column sulphur dioxide ($SO_2$), column nitrogen dioxide ($NO_2$, by Mark IV only), spectral UV radiation, aerosol optical depth, and effective ozone layer temperature (e.g., Bais et al., 1996; Cede et al., 2006; Fioletov et al., 2002; Kerr, 2002; Kerr et al., 1981; Savastiouk, 2006). However, the main data product provided by the Brewer is the total column ozone via direct-sun observations. In this work, we focus on the Brewer direct-sun total column ozone data product only, although total column ozone also can be retrieved

using solar zenith-sky radiance, solar global spectral UV irradiance, and lunar direct irradiance (Fioletov et al., 2011; Kerr, 2010). Brewer data was processed by Brewer Processing Software (BPS) developed by ECCC (Fioletov and Ogyu, 2008). The software demonstrated good performance in a recent comparison of available processing software tools for Brewer total ozone retrievals (Siani et al., 2018).

The Brewer spectrophotometer is a modified Ebert grating spectrometer that was designed to measure almost simultaneously the intensity of radiation at six selected channels in the UV (nominally 303.2, 306.3, 310.1, 313.5, 316.8, and 320.1 nm). The





first channel is almost exclusively used for wavelength calibration. The four longer wavelengths are used for the total column ozone ($\Omega$) retrieval via the following equation:

$$F + \Delta\beta \cdot m = F_0 - \Delta\alpha \cdot \Omega \cdot \mu \qquad (1)$$

where, $m$ is the effective pathlength of direct radiation through air, $\mu$ is the ratio of effective pathlength of direct radiation through ozone to vertical path (also known as the ozone air mass factor). $F$, $\Delta\alpha$, and $\Delta\beta$ are linear combinations of the logarithms of the measured intensity, the effective ozone absorption coefficients, and the Rayleigh scattering coefficients, respectively. For example, $F = log(I_3) - 0.5log(I_4) - 2.2log(I_5) + 1.7log(I_6)$, where $I_3$ to $I_6$ are the measured intensities of radiation at channel number three to number six. $F_0$ is the ETC, which is the instrument response ($F$) if there were no atmosphere between the instrument and the sun. Details about the standard Brewer ozone retrieval algorithm can be found in Kerr (2010) and the references cited there. In the standard Brewer algorithm, $\Delta\beta$, $m$, and $\mu$ are determined and pre-calculated, and are non-instrumental dependent. $F_0$ and $\Delta\alpha$ (calibration constants) are unique for each instrument and dependent on the exact wavelengths and band passes of the slits of each instrument (Kerr et al., 1985). After laboratory and field calibration (to determine $\Delta\alpha$ and $F_0$, respectively), $\Omega$ is then readily calculated for each field observation (i.e., $F$).

As previously described, to maintain the high precision of all Brewer instruments (i.e., transfer the $F_0$ value), the world reference instruments (BrT and BrT-D) receive their $F_0$ values via the independent calibration technique. These values are transferred to the travelling standard and then to the field Brewers via co-located field calibration routines (i.e., calibration transfer method). The primary calibration history of the world Brewer reference instruments is summarized in Table 2. Due to building roof work at the Toronto site, the BrT-D was temporarily moved to Egbert, Canada (44.230° N, -79.780° W) at the beginning of September 2018 and deployed on the roof of the ECCC Centre for Atmospheric Research Experiments building (CARE, 251 m a.s.l.). The CARE building is located in a rural area, which is surrounded by farmlands. For this period between September 2018 and December 2019, the BrT-D was located about 55 km north west from the BrT. This period of data is still used in the analysis to study and illustrate some fine scale variations in the ozone field. More details about reference instruments' repair and upgrade review are provided in the supplementary information.



**Table 2. Independent calibration history of world Brewer reference instruments.**

| Serial no. (Model version) | Operation since | Independent calibration |
|---|---|---|
| No. 008 (Mark II) | 1984 | March 1999 MLO* |
| | | July 2005 MLO |
| | | Instrument failure in July 2007 |
| | | Oct 2008 Izaña |
| | | Oct 2015 MLO |
| No. 014 (Mark II) | 1984 | Apr 2000 MLO |
| | | July 2005 MLO |
| | | Nov 2008 MLO |
| | | Oct 2013 MLO |
| No. 015 (Mark II) | 1984 | Apr 2002 MLO |
| | | Nov 2010 MLO |
| | | Oct 2013 MLO |
| | | Nov 2017 MLO |
| No. 145 (Mark III) | 1998 | Oct 2008 Izaña |
| | | Oct 2015 MLO |
| | | Oct 2019 MLO |
| No. 187 (Mark III) | 2007 | Nov 2010 MLO |
| | | Oct 2015 MLO |
| No. 191 (Mark III) | 2009 | Oct 2013 MLO |
| | | Nov 2017 MLO |

*MLO: Mauna Loa Observatory.

## 2.2 Pandora

The Pandora instrument records spectra between 280 and 530 nm with a resolution of 0.6 nm (Herman et al., 2009, 2015; Tzortziou et al., 2012). It uses a temperature-stabilized Czerny-Turner spectrometer with a $2048 \times 64$ pixels CCD detector. The spectra are analyzed using total optical absorption spectroscopy (TOAS) technique (Cede, 2019), in which absorption 155 cross-sections for multiple atmospheric absorbers such as ozone, $NO_2$, and $SO_2$, are fitted to the spectra. Different from the Brewer, which only uses intensities measured at four wavelengths, the Pandora instruments use the entire spectrum from 310 to 330 nm (at 0.6 nm resolution, with more than 160 pixels) in its ozone retrieval. The current Pandora standard ozone column retrieval algorithm uses a literature reference spectrum (composite of Kurucz (2005), Thuillier et al. (2004), van Hoosier (1996) and Gueymard (2004), details in Cede (2019)), and does not retrieve the effective ozone temperature. Thus, Pandora standard 160 ozone data products have a temperature dependence (Herman et al., 2015), i.e., 0.25 % $K^{-1}$ when compared to Brewer measurements (Zhao et al., 2016). This temperature dependence introduces a 1 to 3 % seasonal bias between the Pandora and the Brewer standard data products. Another major difference between the Brewer and Pandora retrieval algorithms is their



selection of ozone cross-section, i.e., the Brewer uses BP (Bass-Paur) ozone cross-section (at 228.3° K, Bass and Paur, 1985) and the Pandora uses DBM (Daumont, Brion and Malicet) ozone cross-section (at 225° K, Brion et al., 1993, 1998; Daumont

et al., 1992; Serdyuchenko et al., 2014). As a result of temperature dependency and different selection of ozone cross-sections, a two percentage multiplicative bias between the Pandora and Brewer standard ozone column products were found in Zhao et al. (2016). Thus, in this work, the Pandora ozone data are corrected by an empirical method with the ozone-weighted effective temperature (Zhao et al., 2016). The effective temperature was calculated from temperature and ozone profiles provided by ERA-Interim (Dee et al., 2011). In general, after correction, Pandora ozone data agrees with the Brewer measurements within

±0.25 % (for seasonal mean values). An effective ozone temperature retrieval algorithm is under development for the Pandora to minimize its temperature dependence effect (Cede, 2019). Additional information on Pandora calibrations, operation, retrieval algorithms and correction method can be found in Cede (2019), Tzortziou et al., (2012), and Zhao et al., (2016).

Pandora instrument no. 103 has been making direct-sun measurements in Toronto (co-located with BrT and BrT-D) since 2013

(Zhao et al., 2016). The instrument has made almost daily measurements since its deployment, except during a filter upgrade in 2017. The seven years of data (2013-2019) have been re-processed and harmonized by the Pandonia Global Network (PGN) to ensure the high quality of its ozone data product. In this work, only high-quality Pandora ozone data products are used (Pandora level 2 (L2) data product quality flag = 0; Cede, 2019). Originally, Pandora no. 103 was operated in DS mode only and Pandora DS ozone data had a one-minute resolution. Starting in 2018, it was operated in the combination mode (i.e., direct-

sun, zenith-sky, and multi-axis) and Pandora DS ozone data had a five-minute resolution. The Pandora and BrT-D instruments have good stray-light control, and their air mass dependence is comparably low up to 81.6° SZA (Zhao et al., 2016).

### 2.3 Satellites

The BrT's performance was evaluated against the Total Ozone Mapping Spectrometer (TOMS) and reported in Fioletov et al., (2005). With more satellite instruments reporting total ozone columns, here we present a data comparison between the Brewer

reference instruments (BrT and BrT-D) and multiple satellites, including TOMS, NOAA Solar Backscatter Ultraviolet Radiometer-2 (SBUV) series (nos. 11, 14, 16, 17, 18, 19), Ozone Mapping and Profiler Suite (OMPS), Ozone Monitoring Instrument (OMI), and TROPOspheric Monitoring Instrument (TROPOMI).

### 2.2.1 TOMS

There were four TOMS in orbit: on Nimbus-7 satellite launched in 1978, on Meteor-3 in 1991, and on ADEOS and Earth

Probe (EP) in 1996. Total column ozone was derived from incident solar radiation and backscattered ultraviolet sunlight measurements. TOMS total column ozone has been widely used for verification of ground-based measurements (e.g., Fioletov et al., 1999; Kyrö, 1993). Fioletov et al. (1999) reported that about 80 % of the Dobson and Brewer data have standard





deviations of monthly mean difference with TOMS that are less than 2.5 %. The EP/TOMS total ozone data from 1996 to 2005 were used in this work (McPeters et al., 1998).

### 2.2.2 SBUV series

Total column ozone from NOAA SBUV series (nos. 11, 14, 16, 17, 18, 19) is used in this work. Unlike TOMS, OMI or TROPOMI, which provides daily global coverage, the SUBV instruments provide full global coverage approximately bi-weekly. The SBUV ozone column data used in this work is produced by the overpass algorithm to create daily overpass values (Labow et al., 2013). Labow et al. (2013) reported that the total column ozone data from Brewers and SBUVs show an agreement within ± 1 % over 40 years (1970-2010).

### 2.2.3 OMPS Nadir Mapper

The OMPS on the Suomi National Polar-orbiting Partnership (Suomi NPP) satellite was launched in 2011 (Flynn et al., 2014; Kramarova et al., 2014). OMPS includes nadir and limb modules to measure both profile and total column ozone concentrations. In this work, OMPS-NPP L2 Nadir Mapper (NM) Ozone Total Column swath orbital v2.1 data from the OMPS-NM module is used. Flynn et al. (2014) reported that the OMPS column ozone (from an earlier v1) has a bias with other records (e.g., OMTO3) on the order of -3 %.

### 2.2.4 OMI

The OMI instrument on the Earth Observing System Aura satellite was launched in 2004. OMI has two standard data products, OMDOAO3 (J. P. Veefkind et al., 2006) and OMTO3 (Bhartia and Wellemeyer, 2002), which are produced using DOAS and TOMS-like techniques, respectively. The mean difference between the two data products varies from 0 to 9 DU (0-3 %) with latitude and season (Kroon et al., 2008).

### 2.2.5 TROPOMI

TROPOMI, onboard the Copernicus Sentinel-5 Precursor satellite, was launched in 2017. The offline (OFFL v010107) total ozone column data (Garane et al., 2019) are used in this work. Garane et al., (2019) reported that the mean bias and the mean standard deviation of the percentage difference between TROPOMI and Brewer ground-based total ozone column data are within 1 % and 2.5 %, respectively.

### 2.3 MERRA-2 reanalysis data

The second Modern-Era Retrospective analysis for Research and Applications (MERRA-2) is an atmospheric reanalysis from NASA's Global Modeling and Assimilation Office (GMAO). MERRA-2 assimilates partial total column ozone retrievals from





the SBUV series from 1980 to 2004. From October 2004, MERRA-2 assimilates ozone profiles and total column data from the Microwave Limb Sounder (MLS) and the OMI, respectively (Wargan et al., 2017). MERRA-2 column ozone data has been found to be of good quality when compared with satellite and ground-based observations (e.g., Rienecker et al., 2011; Wargan et al., 2017; Zhao et al., 2017, 2019). In this work, the MERRA-2 total column ozone ($0.5° \times 0.625°$, version 5.12.4) with 1-hour temporal resolution is used as an input in the third-party comparison model (see Section 4 for more details).

## 3 Comparison methods

Multiple Brewer instruments at the same site may not measure ozone at exactly the same time. To compare the ozone column data provided by each Brewer reference instrument, a "baseline" ozone column value at the time of each measurement should be established. Ideally, if the true ozone column values are known, then the performance of each instrument can be evaluated as simple as calculating the discrepancies between true ozone and measured ozone. However, this approach is not possible in

reality. Several other means to form these (daily or time-resolved) baseline ozone values were used in the past: 1) the average of all satisfactory measurements for each instrument (Kerr et al., 1998), 2) a second-order time-resolved statistical model (Fioletov et al., 2005), 3) a third-order simple polynomial fit (Stübi et al., 2017), and 4) a fourth-order time-resolved statistical model (León-Luis et al., 2018). In general, these approaches aim to define the best baseline total column ozone values for each day, which are as close to true ozone values as possible. Apparently, the first method (i.e., simple daily mean) is not ideal since

it includes the effects of ozone changes during the day combined with differences in the timing and number of measurements by each instrument (Fioletov et al., 2005) and instrument uncertainties are overestimated. The second method takes the daily baseline ozone values as a second-order function, which are fitted using all satisfactory measurements for all three instruments together, but also give the individual instrument a degree of freedom in offsets. The third method takes the ozone changes into account, but it is still affected by the number of measurements from each instrument (i.e., the instrument reporting more data

points will dominate the baseline). The advantage of the time-resolved model (second or fourth method) is that it takes both effects of ozone changes into account and minimizing the impact of sampling (i.e., all three instruments share the same first and second-order terms, while the offset terms are unique for each instrument; see more details in the following section). It should also be noted that third- or higher-degree polynomial fit does not really change the results much because the baseline is only needed to adjust for the time difference in ozone measurements by individual Brewers. Thus, to make the current work

directly comparable to previously reported results for the world reference instruments, we only use the second approach in the analysis (i.e., second-order time-resolved statistical model; following Fioletov et al., (2015) referred to as Model 1).

In addition to constructing the baseline with the individual Brewers' data, we can use third-party (e.g., co-located, independent total column ozone measurements from Pandora) measurements as the baseline ozone in the evaluation. The Pandora

instrument typically has a better temporal-resolution than Brewers, and therefore, can capture most of the daily ozone variations



better. Moreover, when using Pandora ozone, the baseline will not have the sampling or weighting issues, i.e., the Brewer instrument that reported more data points will not dominate the forming of the baseline. However, when using this third-party baseline, we should be cautious about the difference between Pandora and Brewer ozone data products, i.e., their seasonal and multiplicative bias. Details about how to interpret the third-party assessment results are provided in Section 4.

## 3.1 Comparison with ground-based instruments

### 3.1.1 The original method

Two statistical models have been developed to evaluate Brewer reference instruments' performance by Fioletov et al., (2005). The first model is a time-resolved second-order model (referred to as Model 1) to provide the baseline ozone and applied to the reference triad data from each day:

$$\Omega = A_1 \cdot I_1 + A_2 \cdot I_2 + A_3 \cdot I_3 + B \cdot (t - t_0) + C \cdot (t - t_0)^2 \qquad (2)$$

where, $\Omega$ is an ozone measurement from one of the three Brewers (e.g., BrT, or here with arbitrary serial nos. 1, 2, and 3), $t$ is the corresponding time of the measurement and $t_0$ is the local solar noon time. The $I_1$, $I_2$, and $I_3$ are the indicator functions for each of the three Brewers. For example, if the ozone value $\Omega$ is measured by Brewer no. 1, $I_1$ is set to 1 (and set to 0 for the two other Brewers). The coefficients $A_1$, $A_2$, $A_3$, $B$, and $C$ can then be estimated by the least-squares method. Please note here the ozone values for this day are then represented by three second-order curves, which share the common curvatures ($B$ and $C$ terms), but have a different offset (i.e., $A_1$, $A_2$, and $A_3$). In other words, each instrument formed its own daily time-resolved ozone variations, but these variations are not totally independent from each other (since they share the $B$ and $C$ terms). Then, the average of the three coefficients A = $(A_1 + A_2 + A_3)/3$ is used as the benchmark to evaluate the performance of individual instruments. For example, $(A_1 - A)$ represents the deviation of Brewer no. 1 from the baseline ozone (i.e., corresponding to $\Omega$ = $A + B(t-t_0) + C(t-t_0)^2$).

In general, with contributions from all three instruments, this model removes the diurnal ozone variations relative to the noon ozone value. Meanwhile, the model preserves the instrumental differences as much as possible by assigning different offsets for each baseline (i.e., corresponding to an assumption that there is only an additive bias between Brewers).

For a well-calibrated and well-maintained Brewer instrument, its major uncertainties in derived ozone column data came from two instrument constants assigned to it (i.e, $F_0$ and $\Delta\alpha$). Next, to further break down the uncertainty budgets, Model 2 is designed by combing Eqns. 1 and 2 as:



$$F + \Delta\beta \cdot m = (F_0' + X) - (\Delta\alpha' + Y) \cdot (A + B \cdot (t - t_0) + C \cdot (t - t_0)^2) \cdot \mu \qquad (3)$$

where, $F_0'$ and $\Delta\alpha'$ are the assigned ETC and effective ozone absorption coefficient values. X and Y are the assigned uncertainties to these two instrument constants. Here, the total column ozone amount ($\Omega$) is replaced by the Model 1 defined baseline ozone. Next, X and Y can be estimated for each of the three instruments using the least-squares method for each 3-month season. In general, Model 2 assumes that the baseline ozone provided by Model 1 is the ground-truth (i.e., true ozone values). Thus, the difference of total column ozone between the individual instrument and Model 1 is allocated to the "error" of ETC and effective ozone absorption values.

### 3.1.2 Third-party scheme

The design of Model 2 is based on our assumption of the high quality of Brewer ozone data, i.e., the Brewer-derived baseline ozone (Model 1 ozone) is close to the true ozone. In general, for well-calibrated and well-maintained Brewer instruments, this assumption is valid. For example, if Brewer nos. 1 and 2 are in good condition, but Brewer no. 3 is not, Model 1 will show the discrepancy. Then, we can easily identify the issue and re-calibrate Brewer no. 3. However, if Brewer nos. 1 and 2 are the instruments with larger discrepancies from true ozone and Brewer no. 3 is in good condition, then things will become more complex. In addition, whenever we select three instruments to form a "triad" and use Models 1 and 2 to performing the analysis, we also selected baseline ozone defined by those three instruments. In other words, the Models 1 and 2 analyses applied to BrT and BrT-D cannot reflect their relative difference, i.e., BrT uses BrT's baseline, whereas BrT-D uses BrT-D's baseline. Thus, to better evaluate and compare BrT and BrT-D's performances, we need to use a third-party ozone column data as the baseline. Here, Model 3 is designed as:

$$F + \Delta\beta \cdot m = (F_0' + X) - (\Delta\alpha' + Y) \cdot \Omega_{3rd-party} \cdot \mu \qquad (4)$$

where, the only difference compared to Model 2 is that we replaced the Model 1 defined baseline ozone with a new third-party baseline ozone ($\Omega_{3rd-party}$). The new baseline can be either another co-located and independent ozone column observations (e.g., Pandora ozone data) or reanalysis data (e.g., MERRA-2). Please note here the $\Omega_{3rd-party}$ has to be independent of Brewer reference instruments. For example, they cannot be measurements from another Brewer (e.g., another co-located field Brewer instrument), unless it received its ETC constant via the independent calibration method.

When a third-party baseline ozone exists, it is easy to evaluate the deviation of each Brewer from the baseline ozone. Thus, in this work, when using the third-party baseline ozone, we simply report their absolute and relative differences defined as:





$$\Delta_{abs} = \Omega_{Brewer} - \Omega_{3rd-party} \qquad (5)$$

$$\Delta_{rel} = \frac{\Omega_{Brewer} - \Omega_{3rd-party}}{\frac{(\Omega_{Brewer} + \Omega_{3rd-party})}{2}} \times 100\%. \qquad (6)$$

**3.2 Comparison with satellites**

Regression analyses between Brewer and satellite observations were made by using the following coincident criteria: (1) nearest (in time) measurement that was within $\pm x$ hr of satellite overpass time, (2) closest satellite ground pixel (having a distance ($d$, in km) from the ground pixel centre to the location of the Brewers less than $y$ km). These coincident criteria are summarized in Table 3. Only good quality satellite data are used in the analysis. For example, OMTO3 with only error flag = 0 (good sample) are used.

**Table 3. Satellite comparison criteria.**

| Satellite (product) | Time criteria $\|\Delta t\| \le x$ hr | Spatial criteria $d \le y$ km |
|---|---|---|
| OMI (OMDOAO3) | 1 | 30 |
| OMI (OMTO3) | 1 | 30 |
| SBUV-11 | 2 | 200 |
| SBUV-14 | 2 | 200 |
| SBUV-16 | 2 | 200 |
| SBUV-17 | 2 | 200 |
| SBUV-18 | 2 | 200 |
| SBUV-19 | 2 | 200 |
| OMPS | 2 | 50 |
| TOMS | 2 | 50 |
| TROPOMI | 0.5 | 10 |

**4 Assessment results**

The assessment of the Brewer reference instruments was performed using Models 1, 2, and 3 defined in Section 3. To ensure the assessment is based on good quality data, the data were strictly filtered (i.e., data from single and double spectrometer instruments with reported standard deviation > 3 DU or µ > 3.5 are removed). Using 3 DU (Fioletov et al., 2005) instead of the standard 2.5 DU (Fioletov and Ogyu, 2008) yields more data points and, therefore, more days suitable for comparison, but does not improve the comparison since the additional measurements are the noisiest.



## 4.1 Comparison of ground-based instruments

### 4.1.1 Model 1


To perform Model 1 analysis, additional criteria are applied. A specific day is analyzed with Model 1 only if each of the three instruments has 1) at least ten measurements on that day and 2) at least three measurements in each half-day (defined by local solar noon time) on that day. The Model 1 analysis was done for BrT and BrT-D separately. The deviations of each individual instrument from their baseline are shown in Fig. 1a, which are comparable to the results in Fig. 1 from Fioletov et. al., (2005).

The residuals from Model 1 include some remaining instrument uncertainties, but also some short-term fluctuations in ozone, which are not reflected by the second-degree polynomial model. The uncertainties include the effects of instrument temperature fluctuations and the differences in the characteristics of the neutral density (ND) filters. The $5^{th}$ and $95^{th}$ percentiles of the Model 1 residuals are shown in Fig. 1b, which are comparable to the results in Fig. 2 from Fioletov et al., (2005). The standard deviation of the residuals is about 2.4 DU or 0.72 %. In general, these updated results show that the performance of the BrT in

the last two decades (1999-2019) is comparable to its reported values from 1984 to 2004. The long-term instrument drifts are still typically within ±1 %. The standard deviations ($\sigma$) of the 3-month averages plotted in Fig. 1a are 0.43 %, 0.36 %, and 0.42 % ($\sigma' = 0.40\%$) for Brewers #008, #014, and #015, which are comparable to the reported values from 1984 to 2004 (0.40 %, 0.46%, and 0.39 %). The double triad also shows good long-term stability with the Model 1 analysis, where all measurements are within ±1% compared to its baseline. The standard deviations are 0.44 %, 0.26 %, and 0.33 % ($\sigma' = 0.34\%$) for Brewers.

#145, #187, and #191. From this, assuming that the instrument uncertainties are independent, the standard uncertainty of Brewers ($\delta$) can be estimated as $\sqrt{1.5}\sigma'$, i.e., 0.49 % and 0.42 % for BrT and BrT-D, respectively.

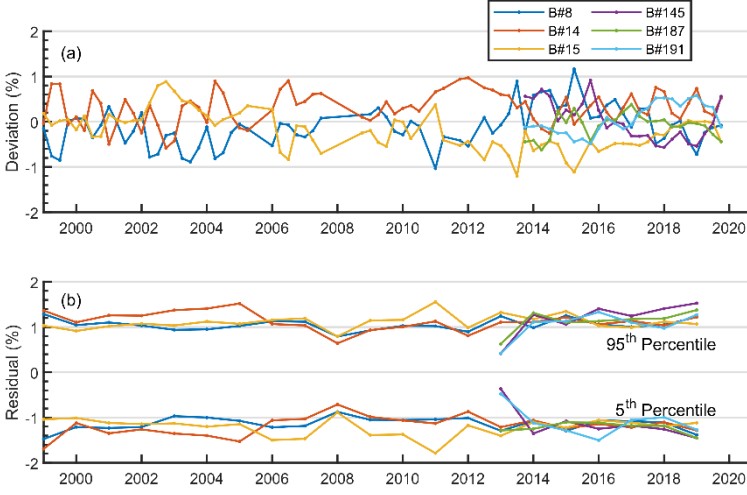

**Figure 1. Model 1 estimated deviations and residual of ozone values. (a) Deviations of ozone values of individual triad Brewers from the mean of the three instruments. Each point on (a) represents a 3-month average. Panel (b) shows the $5^{th}$ and $95^{th}$ percentiles of the residuals of the Model 1 analysis. Each point on (b) is based on one year of data.**






### 4.1.2 Model 2

The Model 2 analysis was performed for BrT and BrT-D. Figure 2 corresponds to Fig. 4 in Fioletov et al., (2005). In general, Fig. 2 shows the errors in the ETCs and effective ozone absorption coefficients account for up to ±2 % of total column ozone, as indicated in Fioletov et al., (2005). Here, the errors in the ETCs and effective ozone absorption coefficients are estimated in R6 ratio units (the units used in the actual Brewer processing algorithm; R6 values corresponding to measured slant column density, i.e., $\Omega = \frac{(R6 - F_0)}{\Delta \alpha \mu}$). The errors are converted from R6 ratio units to percentages of total column ozone by using typical conditions for Brewer measurements in Toronto (i.e., $\Omega$ = 330 DU, $\alpha$ = 0.34, and μ = 2), to provide more straightforward values to assess the impact of errors in the ETCs and effective ozone absorption coefficients. In typical conditions, the uncertainties of ozone absorption coefficient are within ± 1 micrometer step based on the dispersion test, which corresponds to approximately ±0.3 % of total column ozone. For the uncertainties of ETC, the goal is to have it within ± 5 R6 ratio units.

The large errors in ETCs and ozone absorption coefficients may largely compensate for each other and not be evident in the Model 1 analysis. For example, during 2013, there were significant errors in the assigned ETCs and absorption coefficients to #008 that was truly caused by wavelength range limitations of this early model Brewer. A measurement type was added to the schedule of this instrument, that when ran, reached the extent of physical travel of the micrometer causing a 2 nm shift in the measurement from the forward to the backward scan of the micrometer. The Model 2 results show that the BrT-D has a similar performance compared to the BrT since 2013. The errors in ETCs and ozone absorption coefficients from BrT-D (within ±1 %) are even smaller than those from BrT in the most recent period (2017-2019).


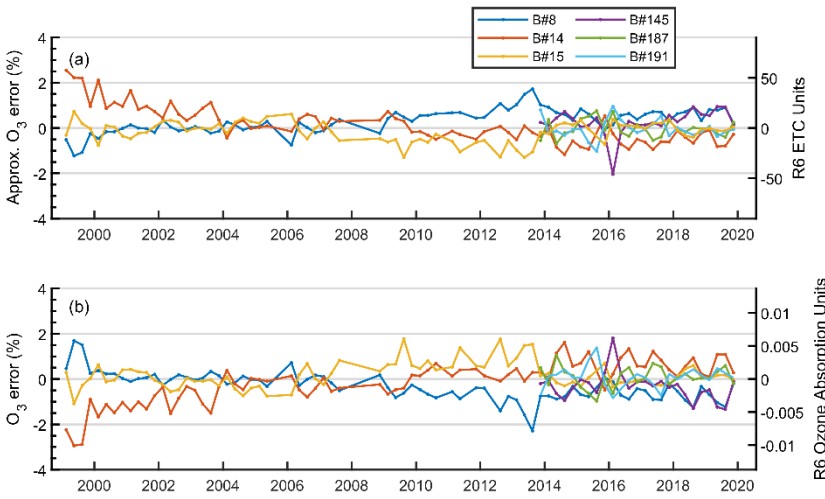

**Figure 2. Relative systematic uncertainties in ETCs and effective ozone absorption coefficients estimated using Model 2. The right y-axes represent the values in the units used in the actual Brewer algorithm (i.e., R6 ratio units); the left y-axes demonstrate the % values of these errors in total ozone values. Each point on the graph represents a 3-month average.**

### 4.1.3 Model 3


For a third-party-based ozone analysis (Model 3), Brewer and Pandora data are both averaged into 10-minute bins and then paired. Note that the Pandora instrument sampling frequencies were reduced from one measurement every 1.5 minutes in 2013-2017 to one measurement every 5 minutes in 2018-2019 due to change in the observation schedules.

Differences between the Pandora observations and the measurements by individual Brewers are shown in Fig. 3. The gaps in the Pandora record are caused by an instrumental failure in winter 2014, Pandora filter wheel upgrade in winter 2017 and persistent cloudy conditions in winter 2018. The absolute differences between Brewer and Pandora data are within ±10 DU. They are slightly larger in wintertime due to the temperature dependency in Pandora ozone data (although empirical correction methods have been applied, the residual effect still exists, e.g., Fig. 13 in Zhao et al., (2016)). For example, the absolute differences from the six Brewer instruments all shifted towards positive in the January to February 2017 period. Thus, when using Pandora data as a third-party baseline, it is more important to examine the variation of relative differences (i.e., $\Delta_{rel}$ of one Brewer minus $\Delta_{rel}$ of another Brewer). In the period of the example, the relative differences between Brewer #015 and Brewer #145 are within 5 DU. Thus, the Brewers' performance was, in fact, stable in that period. Figure 3 shows the relative differences, indicating that compared to Pandora, all Brewer reference instruments have long-term stability within ±2 %. This result is not as good as the prediction from Model 1 (which shows ±1 % deviations) because even corrected, Pandora data still



have some residual seasonal bias. For shorter periods (e.g., summer 2016), all six Brewers have a relative difference within
the range from 0 to -2 %, which is comparable to a ±1 % when Brewer themselves are used as baselines.

We also can assess the performance of individual instruments from a third-party ozone baseline. For example, when compared
to any other reference instruments, Brewer #015 gave the lowest ozone in the period from 2015 to 2017. Another example is

the period after the BrT-D was relocated to Egbert, in which the discrepancy between BrT and BrT-D data became obvious
(up to 4 % relative difference between Brewers #008 and #191).

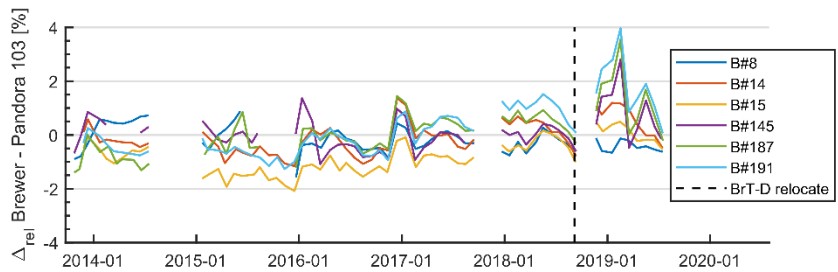

**Figure 3. Monthly relative differences between Brewers and Pandora total column ozone. Monthly averages are calculated if there**
**are at least ten coincident measurements between Brewer and Pandora for that period. The black dash line represents the time when**
**BrT-D was relocated to Egbert, i.e., Pandora and BrT-D were not co-located.**

The Model 3 analysis results are shown in Fig. 4, where the errors in ETCs and ozone absorption coefficients from each Brewer
are reported independently. They show that the quality of these instrument "constants" can drift in time due to the nature of

the calibration and maintenance work performed on the instruments. In general, Fig. 4 shows that in most cases, the estimated
ETC and effective ozone absorption errors for all reference instruments are within ±2 %, i.e., similar to the Model 2 results
(see Fig. 2).





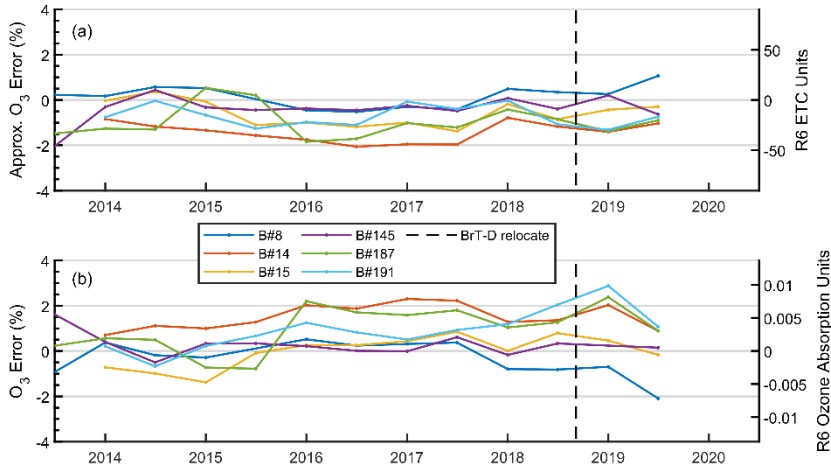

**Figure 4. Relative systematic uncertainties in ETCs and effective ozone absorption coefficients estimated using Model 3. Description of y-axes is in Fig. 2. Each point on the graph represents a 6-month average. The black dash line represents the time when BrT-D was relocated to Egbert.**

When compared to Model 2, Model 3 provides independent estimates of ETC and effective ozone absorption errors, i.e., errors for BrT and BrT-D can be compared directly. For example, in Fig. 2a, we cannot directly compare the ETC errors from Brewer #014 with those from Brewer #145 because they were evaluated by different baselines. However, with Fig. 4, we can conclude that Brewer #145 has about 1 % lower ETC errors than those for Brewer #014. The detailed results of ETC and ozone absorption coefficients errors are summarized in Table 4. In general, for this assessment period (2013-2019), Brewers #008 and #145 have lower ETC and effective ozone absorption coefficients errors (within ±0.5 %) when compared to the other Brewer reference instruments.

**Table 4. Mean errors of $\Delta\alpha$ and ETC for Brewer reference instruments (2013-2019).**

| Brewer serial no. | Mean error of $\Delta\alpha$ [R6 absorption unit] | Mean error of ETC [R6 ETC unit] | Mean error of $\Delta\alpha$* [%] | Mean ETC-related error[#] [%] |
|---|---|---|---|---|
| #008 | -0.0011 | 3.69 | -0.33 | 0.16 |
| #014 | 0.0052 | -32.12 | 1.50 | -1.42 |
| #015 | -0.0001 | -13.51 | -0.02 | -0.60 |
| #145 | 0.0010 | -8.70 | 0.27 | -0.39 |
| #187 | 0.0033 | -22.00 | 0.97 | -0.97 |
| #191 | 0.0032 | -15.90 | 0.93 | -0.70 |

* Mean % error in total column ozone, related to error in ozone absorptions; [#] Mean % error in total column ozone, related to error in ETC, corresponding to X when $\mu$ = 2 (see Eqn. 3).





As discussed in Section 4.1.2, sometimes, Model 2 may also overlook issues if two out of three instruments have the compensation effect (i.e., errors in ETCs and ozone absorption coefficients compensate each other). For example, when analyzing Brewer #145 data, it was revealed by the Model 3 analysis that its absorption coefficients were not ideal. The issue was not observed with Model 2 due to Brewer #191 also has a similar issue in the same period. Thus, besides providing independent uncertainties, the Model 3 analysis can provide an important additional quality control process. Details about this additional quality control process are provided in Appendix B.

## 4.2 Comparison with satellite and reanalysis data

Eleven satellite overpass column ozone datasets are used for data verification of the Brewer reference instruments. Figure 5 shows the relative differences between satellite and Brewer measurements for seasonal (3 months) values are within ±4 % in these two decades (1999-2019). The standard deviation ($\sigma_{3month}$) of the 3-month Brewer-satellite relative differences is 1.38 %. Detailed regression analysis was also performed and some results are summarized in Fig. 6.

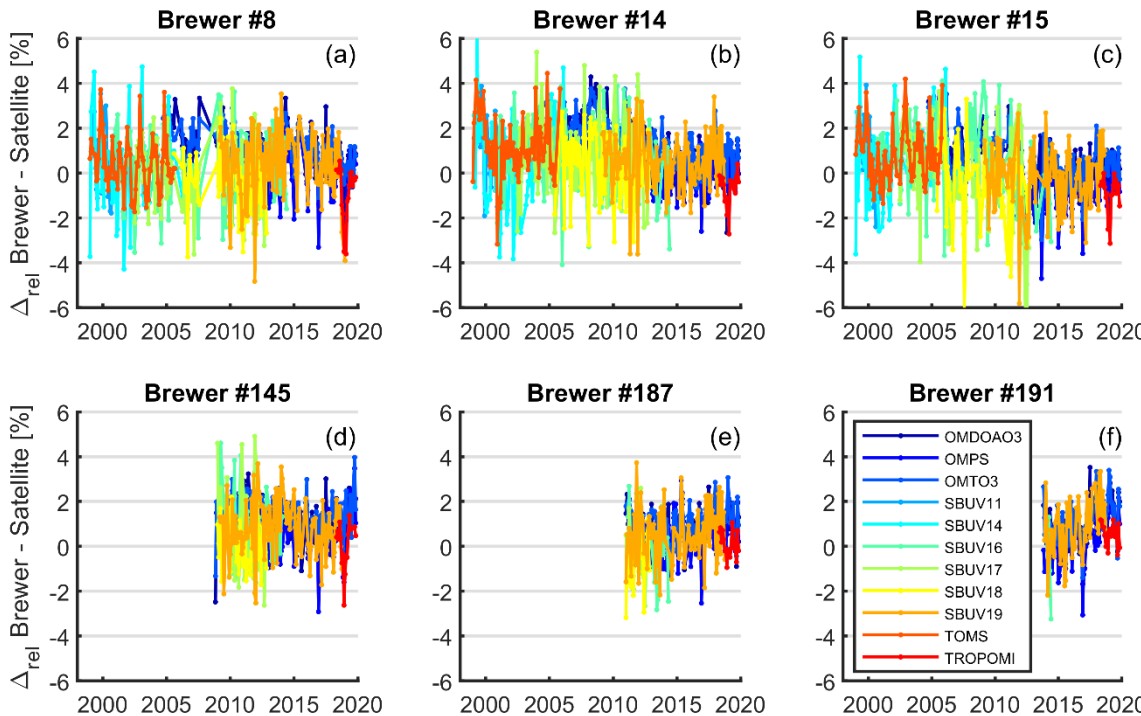

**Figure 5. The relative difference between satellites and the world Brewer reference triads (BrT and BrT-D). Each point represents a 3-month average. Brewers and satellite data are paired with the criteria shown in Table 3.**



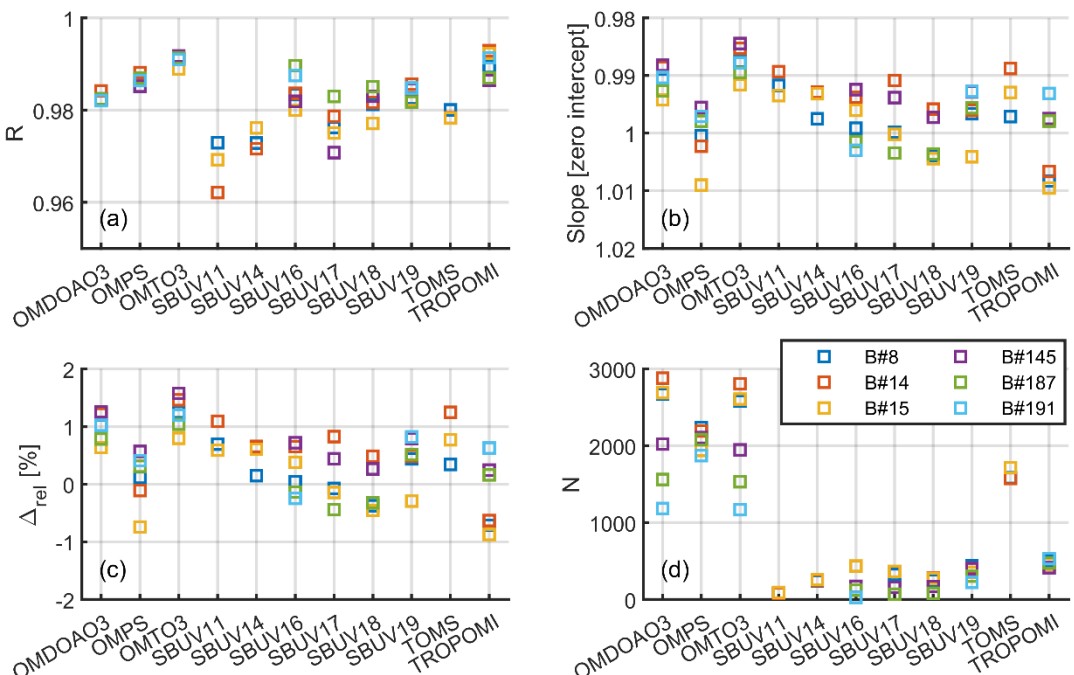

**Figure 6. Summary of the regression analysis between satellites and the world Brewer reference triads. The four panels represent the (a) correlation coefficient (R) between individual Brewer instruments and different satellites (labelled at the bottom axis), (b) the slope of the zero intercept regression line (multiplicative bias), (c) relative percentage difference (bias), and (d) the total number of coincident observations.**

In general, the measurements from the individual Brewers have -1 to 2 % relative difference when compared with all these eleven satellite datasets, with correlation coefficients > 0.96. For most satellite datasets, the regression with zero intercept (Fig. 6b) also shows that the multiplicative bias between Brewers and satellites are well within ±1 %. It is known that satellite data also have some biases and drifts (e.g., Antón et al., 2009; Kroon et al., 2008); therefore, the Brewer-satellite difference values alone do not represent the Brewer instrument performance. Comparison with OMI (both versions) shows that besides the 1 % systematic difference between Brewers and satellite data, the spread of biases with individual instruments is also around 1 %. The standard deviation of the Brewer-OMTO3 (OMDOAO3) difference (for 3-month averages) calculated for six instruments is 0.99 % (1.06 %), about 0.5 % higher than Brewers' standard random uncertainties calculated in Section 4.1.1. In general, BrT and BrT-D's stabilities are assessed by using each satellite dataset, via the standard deviations of 3-month Brewer-satellite relative differences, as shown in Table 5. The results show that the BrT-D ($\sigma_{3month}$ = 1.15 %) has a slightly better long-term stability than the BrT ($\sigma_{3month}$ = 1.33 %), which is consistent with the results in Section 4.1.1 that BrT-D has lower random uncertainty than BrT.





**Table 5. Mean ($\Delta_{rel}{'}$) and standard deviation ($\sigma_{3month}$) of the 3-month Brewer-satellite relative differences.**

| Satellite Dataset | BrT $\Delta_{rel}{'}$ [%] | BrT $\sigma_{3month}$ [%] | BrT-D $\Delta_{rel}{'}$ [%] | BrT-D $\sigma_{3month}$ [%] |
|---|---|---|---|---|
| OMDOAO3 | 0.84 | 1.17 | 0.95 | 0.86 |
| OMPS | -0.30 | 1.07 | 0.39 | 0.96 |
| OMTO3 | 1.14 | 1.08 | 1.30 | 0.80 |
| SBUV11 | 0.93 | 1.59 | N/A | N/A |
| SBUV14 | 0.42 | 1.76 | N/A | N/A |
| SBUV16 | 0.38 | 1.59 | 0.40 | 1.60 |
| SBUV17 | 0.26 | 1.71 | 0.23 | 1.75 |
| SBUV18 | -0.09 | 1.60 | 0.05 | 1.24 |
| SBUV19 | 0.21 | 1.45 | 0.69 | 1.26 |
| TOMS | 0.82 | 1.28 | N/A | N/A |
| TROPOMI[*] | -0.84 | 0.95 | 0.27 | 0.73 |
| **Mean[#]** | 0.34 | 1.33 | 0.54 | 1.15 |

[*] The comparison includes period when BrT and BrT-D were not collocated (see Section 5 for more details). [#] Mean: Only include the satellite datasets that have overlap with both BrT and BrT-D. N/A: not applicable.

To compare with the hourly reanalysis data (MERRA-2 column ozone for Toronto), Brewer column ozone data were resampled to hourly mean values. The relative difference in time series is shown in Fig. 7, which demonstrated the same long-term stability of the Brewer reference instruments when compared with Pandora or satellite instruments. For example, same as Fig. 3 (comparison with Pandora), Brewer #015 is found to have the lowest column ozone values from 2015 to 2018. In general, the relative differences between Brewers and the reanalysis datasets are within ±2 %. The inter-instrument differences (i.e.,

the differences between Brewers) are within ±1 % for most of the measurement period.

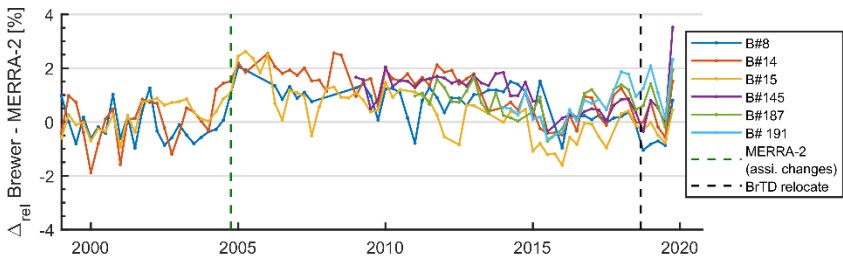

**Figure 7. The relative difference between the reference Brewers and MERRA-2 reanalysis. Each point represents a 3-month average. The green dash line represents the time when MERRA-2 changed its assimilation sources from SBUV-2 to MLS/OMI. The black dash line represents the time when BrT-D was relocated to Egbert.**





The shift in relative difference found in 2004 was due to MERRA-2 changing its data assimilation sources (see the green dash line in Fig. 7). MERRA-2 assimilates partial column ozone data from SBUV instruments between January 1980 and September 2004. Starting from October 2004, MERRA-2 assimilates ozone profiles and columns from MLS and OMI instruments

(Wargan et al., 2017). For example, the mean Brewer #014 – MERRA-2 relative bias was 0.11 % ($\Delta_{rel}{'}$) for the SBUV-based data assimilation, but it increased to 1.07 % after October 2014, probably due to some bias in OMI data as mentioned previously in Section 4.2. For the MLS/OMI-based assimilation period, the multiplicative biases between individual Brewer instruments and MERRA-2 are from 0.40 % (for Brewer #015) to 1.05 % (for Brewer #014); therefore, the relative biases between Brewers are within 0.65 %. In addition, the standard deviation of the 1-month percentage difference is on average 1.04 % for BrT and

0.87 % for BrT-D. Details of the comparison between Brewer reference instruments and the MERRA-2 reanalysis ozone dataset are summarized in Table 6.

**Table 6. Brewer reference instruments vs. MERRA-2 reanalysis ozone dataset.**

| Brewer serial no. | SBUV-based [1999 – Sep. 2004] | | | MLS/OMI-based [Oct. 2004 - 2019] | | |
|---|---|---|---|---|---|---|
| | $\Delta_{rel}{'}$ [%] | M-Bias[*] [%] | $\sigma_{1month}$ [%] | $\Delta_{rel}{'}$ [%] | M-Bias[*] [%] | $\sigma_{1month}$ [%] |
| #008 | -0.13 | -0.27 | 1.14 | 0.61 | 0.69 | 0.98 |
| #014 | 0.11 | 0.16 | 1.20 | 1.07 | 1.05 | 1.04 |
| #015 | 0.21 | 0.18 | 1.12 | 0.39 | 0.40 | 1.11 |
| #145 | N/A | N/A | N/A | 1.01 | 1.02 | 0.89 |
| #187 | N/A | N/A | N/A | 0.79 | 0.71 | 0.81 |
| #191 | N/A | N/A | N/A | 0.76 | 0.66 | 0.92 |

[*]Multiplicative bias is estimated with the slope of zero intercept linear regression. N/A: not applicable.

**5 Discussion**

The performance of the European regional reference instruments (i.e., RBCC-E triad) was reported by León-Luis et al., (2018) and compared with the world reference instruments, specifically the BrT. León-Luis et al., (2018) reported that RBCC-E instruments have a mean 3-month standard deviation ($\delta_{3month}$) of 0.27 %, and concluded that the RBCC-E instruments have 36 % lower $\delta_{3month}$ when comparing to the world reference instruments (i.e., BrT, 1984-2004 period, $\delta_{3month}$ = 0.39 %). However,

the comparison was not straightforward. The Model 1 analysis carried out in León-Luis et al., (2018) did not follow the Model 1 design described in Fioletov et al., (2005) and the current work. It is worth noting that the baseline ozone should be the same (except for the offset) for all three RBCC-E instruments. This would be achieved by including the indicator functions described in Section 3.1.1. The 3-month standard deviations of the BrT, BrT-D and RBCC-E instruments (with corresponding data periods) are summarized in Table 7; however, the results from the RBCC-E instruments should not be directly compared to





the ones in Fioletov et al., (2005) or the current work. Moreover, Stübi et al., (2017) examined three Brewer instruments located at Arosa and found a similar performance of short-term variability. They reported that the standard deviation of short-term variability of the Arosa Brewer triad since 1998 was estimated to be about 0.36 % on the scale of a decade. The medium-to long-term stability was estimated to be within ±0.5 %.

**Table 7. World and European regional reference instruments' 3-month standard deviations.**

| BrT 1999-2019 (1984-2004) | | BrT-D 2013-2019 | | RBCC-E 2005-2016 | |
|---|---|---|---|---|---|
| Serial no. | $\sigma_{3month}$ [%] | Serial no. | $\sigma_{3month}$ [%] | Serial no. | $\sigma_{3month}^{*}$ [%] |
| #008 | 0.43 (0.40) | #145 | 0.44 | #157 | 0.29 |
| #014 | 0.36 (0.46) | #187 | 0.26 | #183 | 0.31 |
| #015 | 0.42 (0.39) | #191 | 0.33 | #185 | 0.20 |

*Calculated with a different method.

It is, however, important to understand that there are certain limitations in the Brewer hardware, which explain why the stability below 0.5 % is so difficult to achieve and maintain. For example, it was found that Brewer #015 has a particularly strong
temperature dependence where the optical frame was expanding faster than its pushrod, an aluminum rod that joins the micrometer to the grating arm that is in place to compensate for normal temperature expansion and contraction. As a result, the spectrum was drifting with temperature faster than was compensated for by the mercury bulb tests, leading to greater than normal variability. This issue was fixed in 2017 by replacing the optical frame. A second example is the original configuration of Brewer #145 micrometer was found to have developed wear and became unreliable, causing some wavelength drifts, and
as a result, relatively high uncertainties for Brewer #145 as shown in Table 7. The top and bottom micrometers were fully replaced in 2019, including all the connecting wires of the wire micrometer system.

Another example of hardware-related issues with Brewer ozone measurements is the characteristics of the ND filters used to reduce the intensity of incoming radiation (Kerr, 2010). In practice, the filters are not always neutral, but may have some wavelength dependence on their transmittance. The Brewer retrieval algorithm removes effects that are linear as a function of
the wavelength, but this offset may not be enough in some cases and a shift of up to a few DU in the retrieved ozone values can occur as a result of a ND filter switch (e.g., from ND filter #1 in the early morning to ND filter #4 in the noon; Savastiouk, 2006). Instruments with ND filters from the same manufactured batch will demonstrate the same shifts in ozone values. Thus, these instruments may have very similar characteristics, and therefore, demonstrate high precision; however, they all may be
affected by the same or similar hardware-related systematic errors. There are other hardware-related factors that affect the accuracy and precision of Brewer measurements. For example, a simple replacement of the mercury bulb that is used to ensure the instrument stability could affect total ozone measurements, creating jumps in the data record. The bulb change has the



potential to affect the CalStep (calibration step, the optimal micrometer position found in the "Sun Scan" test; Savastiouk, 2006) of the instrument. If the combined focus of the monochromator mirrors of the instrument is not optimized and the

illuminated filament of the mercury bulb is located in a significantly different location than the illuminated filament from the original bulb, as much as a 5 micrometer step (one micrometer step is 0.7 pm) change may be seen. It is best to change the mercury bulb before it completely fails so that sequential mercury tests can be performed using both bulbs to detect and address any shifts in the CalStep. It is still recommended to perform the "sun scan" test and verify any potential changes.

The way that the data are processed also affects the results. Siani et al, (2018) concluded that the ozone data processed by different software agree at the 1 % level; however, some differences can be found depending on the software in use. They also recommended "a rigorous manual data inspection" of the processed data and to be careful with how Standard Lamp (SL) test results are used. Visual data screening was also used by Stübi et al., (2017) to eliminate outliers. However, this approach raises the question of reproducibility of the obtained results and must be carefully documented.


Validation of satellite data is an important application of Brewer measurements and the modern satellite instruments demonstrated agreement with Brewers within 1 % (e.g. Garane et al., 2019). At the 1 % level, there are many factors that affect the comparison results. Some of the factors related to ozone absorption cross-sections and their temperature dependence are well established (e.g., Redondas et al., 2014). However, the high spatial resolution of modern satellite instruments such as

TROPOMI brings new challenges. Figure 8 shows that TROPOMI overpass (OVP) data from the Downsview site in Toronto (centre of ground pixels within 10 km from Downsview) have a better agreement with those of the BrT-D when it was relocated to Egbert than with those of the Brewers at Toronto. The difference is about 2 %, which is too large to be explained by, for example, stray light. It is likely related to a difference in viewing geometry. For a Brewer, the light passes through the ozone layer once along the line between the instrument and the sun; for a satellite measurement, the light passes through the ozone

layer in the same way as for ground-based measurements, but then is backscattered by the atmosphere and surface toward the satellite sensor and passes through the ozone layer again. In the case of a large latitudinal gradient, the thickness of the ozone layer could be very different (Fig. 8b). As shown by the green and purple lines, the Downsview Brewers were sampling stratospheric ozone over Hamilton, while the Egbert Brewers were sampling stratospheric ozone over Brampton. The previous generations of satellite instruments had spatial resolution in the order of $50 \times 50$ km$^2$ (except for OMI) and the difference in

the viewing geometry had only a minor impact. However, for current and future high-resolution satellites, such as TROPOMI and TEMPO (Zoogman et al., 2014), these sampling effects should be taken into account for future satellite ozone validation works (e.g., Verhoelst et al., 2015). In general, we conclude that all these reference instruments show good long-term stability as well as meet the WMO/GAW requirements.





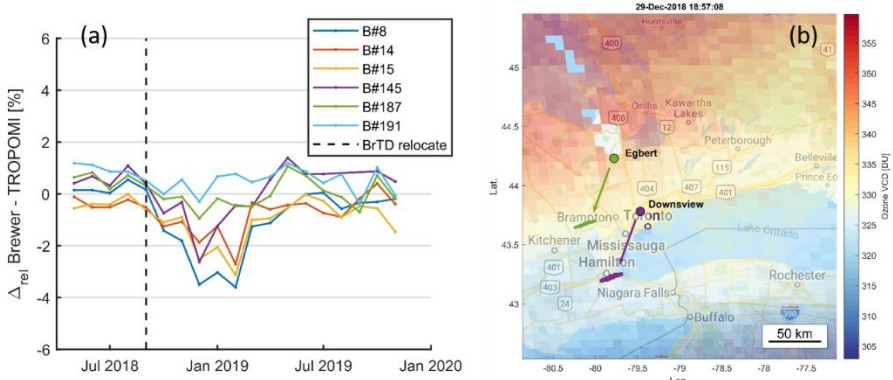

**Figure 8. Example of small scale column ozone field variation. (a) Monthly relative differences between Brewers and TROPOMI total column ozone overpass measurements (for Downsview in Toronto) and (b) TROPOMI total column ozone measured on 29 December 2018 over southern Ontario, masked with Brewers' viewing directions and sampling areas. The base map is from © Google Maps.**

## 6 Conclusion

This work assessed the long-term performance of the world Brewer reference instruments, maintained by ECCC in Toronto, Canada, in measuring total column ozone. The last assessment of the BrT was done in 2005 with two decades of ozone data records from 1984 to 2004. This work provides a more recent assessment for the BrT (1999-2019) and reports the first assessment of the BrT-D (2013-2019). It was found that both single and double reference triads met the WMO/GAW ozone monitoring requirements. Using statistical models, both BrT and BrT-D have a better than 0.5 % precision. The 3-month

standard deviation of ozone values from the two triads are well within 0.5 %, with BrT-D having slightly better performance (BrT and BrT-D have mean standard deviations of 0.40 % and 0.34 %, respectively). In addition, the BrT-D has proven to have better performance in low sun conditions (see Appendix A), which provides benefits in ozone monitoring work in the Polar Regions. Comparison with Pandora total ozone measurements (adjusted for temperature dependence) re-confirmed the high quality of the world Brewer reference instruments. It was found that both BrT and BrT-D have a difference of less than

570    0.5 %.

Further detailed error budget analysis shows the impacts of ETC and ozone absorption coefficients errors for both reference triads are within ±2 % when the statistical Model 2 is used. This result is comparable to the BrT findings for data records from 1984 to 2004. When using the Pandora as a reference (Model 3), the ETC and ozone absorption errors from BrT-D are slightly

better than the ones from BrT (±1.5 % and ±2.0 % for BrT-D and BrT, respectively). It demonstrates that all reference instruments were well-calibrated and maintained in good condition.



Differences between the measurements from the individual Brewer triad instruments and eleven satellite datasets are within -1 to +2 %. For most satellite datasets, the multiplicative bias between Brewers and satellites is well within ±1 %. The viewing
geometry (or line-of-sight) of ground-based and satellite instruments should be considered in future high-resolution satellite ozone validation activities. Moreover, a 20-year long-term reanalysis data was compared with the reference Brewers' data record. It shows that the reanalysis data has good quality, with the relative difference between the reference Brewer and the reanalysis datasets being within ±2 %. However, the changing of assimilation sources will affect the quality of the reanalysis and should be addressed in any ozone trend analysis.


The uncertainties of the Brewer triad instruments are under 0.5 %, while the differences with the best satellite instruments and reanalysis data are close to or slightly lower than 1 %. Further improvement of Brewer total ozone observation precision may be limited by the present Brewer 5-wavelength algorithm and Brewer hardware itself. If highly precise Brewer total ozone measurements are required, then the "group-scan" algorithm (Kerr, 2002) that can deliver measurement uncertainties of
individual measurements as low as 0.5 DU or 0.15 to 0.2 %, should be considered instead of the present 5-wavelength method.

*Data availability.* Brewer data are available from WOUDC (https://woudc.org/, last accessed: June 2020). Pandora data are
available from the Pandonia Global Network (http://data.pandonia-global-network.org/, last accessed: June 2020). SBUV data are available from https://acd-ext.gsfc.nasa.gov/anonftp/toms/sbuv/AGGREGATED/, last accessed: June 2020. OMI data are available from https://gs614-avdc1-pz.gsfc.nasa.gov, last accessed: June 2020. OMPS-NM data are available from doi:10.5067/0WF4HAAZ0VHK, last accessed: June 2020. MERRA-2 data are available from doi:10.5067/VJAFPLI1CSIV, last accessed: June 2020. Any additional data may be obtained from Xiaoyi Zhao (xiaoyi.zhao@canada.ca).


*Author contributions.* XZ analyzed the data and prepared the manuscript, with significant conceptual input from VF, and critical feedback from all co-authors. MB, RS, AO, VF and SCL operated and managed the Canadian Brewer Spectrophotometer Network. IA, MB, AO, VF and XZ performed all Brewer triads' data preparation. VS and MB performed independent calibrations for the Canadian Brewers. JD, XZ, VF and SCL operated and managed the Canadian Pandora
measurement program. AC, MT, and MM from the Pandonia Global Network team provided critical technical support to the Canadian Pandora measurement program. MM and MT performed the Pandora ozone harmonization work for the Downsview site in Toronto. DG and CM provided TROPOMI ozone data and supported satellite comparison.





***Acknowledgements.*** We gratefully acknowledge the National Oceanic and Atmospheric Administration (NOAA) and Mauna
Loa Observatory staff for their support in Brewer calibrations. We thank the State Meteorological Agency of Spain
(AEMET) group for hosting Brewer #008 and #145 in the 2008 calibration campaign in Izaña. We thank Nader Abuhassan,
Daniel Santana Diaz, Manuel Gebetsberger, and others from SciGlob and the Pandonia Global Network (PGN) for their
technical support of Canadian Pandora measurements. The PGN is a bilateral project supported with funding from the National
Aeronautics and Space Administration (NASA) and the European Space Agency (ESA). We acknowledge NASA Earth
Science Division for providing OMITO3 data. The Sentinel 5 Precursor TROPOMI Level 2 product is developed with funding
from the Netherlands Space Office (NSO) and processed with funding from ESA. We thank NOAA for providing NOAA
SBUV data. We acknowledge the Goddard Earth Sciences Data and Information Services Center (GES DISC) for providing
OMPS data. We thank the Global Modeling and Assimilation Office (GMAO) for providing MERRA-2 data.

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



**Appendices**

### A. Distribution of standard deviations of individual DS measurements

Figure A1 shows the distribution of the measurement standard deviation ($\delta_M$), which is used to determine the acceptability of each DS ozone data point in the Brewer data processing algorithm. For Brewers, each final DS ozone data point is a mean of five individual measurements (performed within 3 minutes), and the $\delta_M$ is the standard deviation of these five measurements. Typically, the total column ozone values are assumed to be stable within the time of these five measurements. Thus, any DS ozone data with $\delta_M > 3$ DU will be removed. Figure 3a in Fioletov et al., (2005) shows the distribution of $\delta_M$ for BrT with $\mu \leq 3.25$. Since the $\delta_M$ is proportional to the measured quality $F$ divided by $\mu$, the variability of $F$ (among five measured $F$) is also influenced by $\mu$. For example, in the $1.00 \leq \mu \leq 1.25$ range, $\delta_M$ of BrT has a peak value of about 1.8 DU. However, in a higher range of $2.75 \leq \mu \leq 3.25$, $\delta_M$ of BrT has a peak value of about 1 DU.

Typically, Brewer DS ozone data are reported only when $\mu \leq 3.5$ (Note: Except this section, all Brewer DS ozone data used in this study have $\mu \leq 3.5$). This is because, for single spectrometer Brewers, measurements at high $\mu$ values are strongly affected by the stray light (Bais et al., 1996; Fioletov et al., 2000; Wardle et al., 1996). The double Brewers were designed to have low stray light (i.e., internal stray light fraction of $10^{-7}$ and $10^{-5}$ for double and single Brewers, respectively) and showed good performance when $\mu > 3.5$ (e.g., Zhao et al., 2016). To demonstrate the benefits of low stray light in double Brewer instruments and make a direct comparison between BrT and BrT-D, the $\mu$ range is extended to higher values ($\mu \leq 4.75$) in this analysis. Figure A1 shows that for typical $\mu \leq 3.25$ conditions, BrT-D has similar performance as BrT. Whereas, for low solar zenith angle (SZA) conditions (e.g., $4.25 \leq \mu \leq 4.75$), double Brewers still have similar distributions at moderate SZA conditions. Please note that since BrT only reports ozone data with $\mu \leq 3.5$, to make sure the comparison and assessment provided in this work is comparable to Fioletov et al., (2005), both BrT and BrT-D data used in any other sections are filtered with the $\mu \leq 3.5$ criteria. However, the capability of measuring ozone value in low sun conditions is very important for the ozone monitoring in Polar Regions where the SZA is large in early springtime. This stray light effect is further illustrated in Fig. A2, in which the percentage difference between Pandora and BrT(BrT-D) are binned by ozone air mass factors. Figure A2 indicates that in low air mass conditions (AMF < 3.5), BrT and BrT-D have similar air mass dependence, which is consistent with the results reported by Tzortziou et al. (2012) and Zhao et al. (2016). Note that Fig. A2 is similar to Figure 15 in Zhao et al. (2016), but with an extended dataset (2013-2015 in Zhao et al. (2016), 2013-2019 in this work). It is found that the air mass dependencies of BrT and BrT-D are consistent within these two periods.

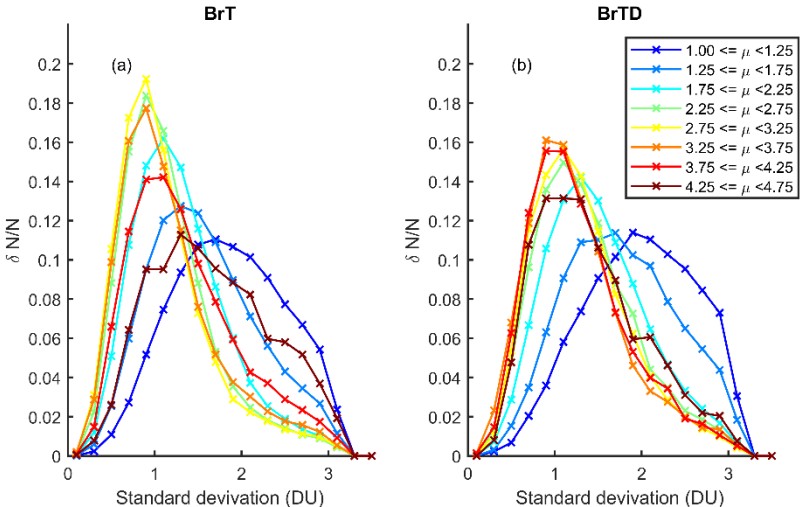

**Figure A1. The distribution of the standard deviations of individual DS measurements as a function of air mass value. (a) shows the Brewer reference triad (BrT) data (1999-2019), (b) shows the double Brewer reference triad (BrT-D) data (2013-2019). Data from all three Brewers for each triad were used for this plot.**

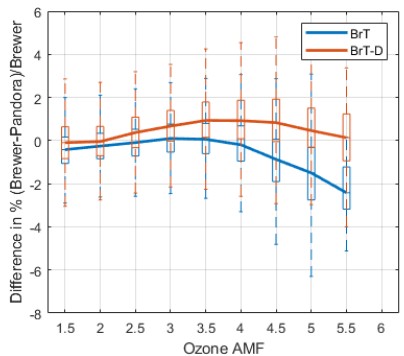

**Figure A2. The percentage difference between Pandora and Brewers (grouped as BrT and BrT-D) as a function of ozone air mass factor. On each box, the central mark is the median, the edges of the box are the 25th and 75th percentiles, and the whiskers extend to the most extreme data points not consider outliers.**





## B. Model 3 analysis improvement examples

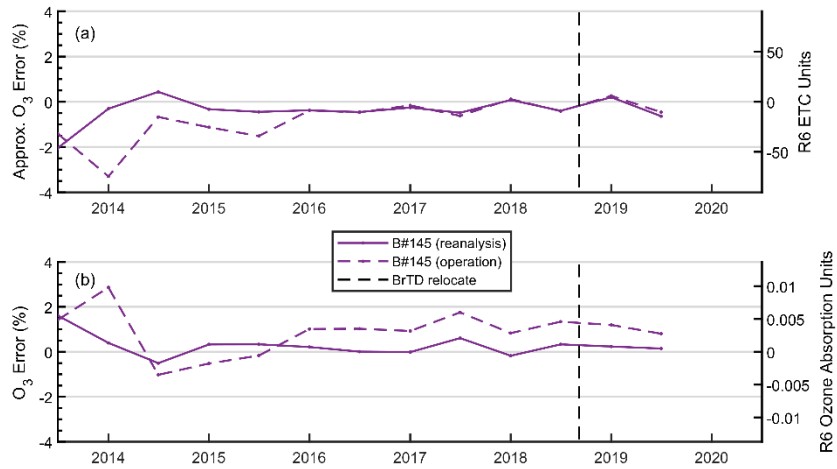

**Figure B1. Comparison between reprocessed and operational data from Brewer #145. (a) Relative systematic uncertainties in ETCs and (b) ozone absorption coefficients estimated using Model 3. Discerption of y-axes in Fig. 2. Each point on the graph represents a 6-month average. The black dash line represents the time when BrT-D was relocated to Egbert.**

815

The early operational processing run of the Brewer triad data, when reviewed through Model 3, indicated that there were some errors in the ETC and absorption values, but were compensating each other when ozone values were calculated. As a result, the used configuration produced a reasonable daily average ozone, but not individual values. For example, Fig. B1 shows that the ETC error in early 2014 was as large as 4 % and the ozone absorption error was about 3 % in the operational processing

820 version. After this observation, the data were scrutinized to find that a calibration step had inadvertently been changed by 5 steps from what was intended. An artificial offset in ozone absorption was introduced in an equal offset to the change in the calibration step to correct for this error. The solid line in Fig. B1 indicates the improvement made.