# Peer review of "The world Brewer reference triad – updated performance assessment"

_Atmospheric Measurement Techniques, 2020_

## Referee Comment (RC1) · Anonymous Referee #1 · 5 Oct 2020

This is the first review of the paper "The world Brewer reference triad – updated performance assessment and new double triad by Xiaoyi Zhao et al. This paper is of great importance for the WMO Brewer network as it discusses the stability of the world Brewer triad maintained by the ECCC, Canada. Comparisons between the single Brewer triad (BrT) and the double Brewer triad (BrT-D) are reported for the 1999-2019 period. The previous assessment of the BrT performance (Fioletov et al., 2005) is used to verify the stability of the reference instruments over an extended period (1984-2019). Four statistical methods to evaluate the uncertainty of each instrument relative to the BrT and BrT-D baseline, to the independent reference observations (Pandora and eleven satellite records), and to the reanalyses (MERRA-2) are presented and summarized in

plots and tables.

The paper is well written, the figures used to demonstrate the analyses are clear. The summary tables support the discussion and allow us to evaluate stability and random uncertainties of the total ozone observation originating from uncertainties in the extra-terrestrial constant (ETC) and the effective absorption cross-section coefficients specific to each instrument in the triad. There are a couple of inconsistencies in the analyses, including grouping of the data in either monthly, 3-months, or 6-months averages. It is not clear why the time periods for averages are changing depending on the analyses. It would make sense to present all data as monthly averages. The 2005 paper analyzed data starting in 1984. Why does this paper exclude the 1984-1988 period? Since the triad is independently calibrated at Mauna-Loa observatory, where the station Dobson (since 1957) is located, why not to perform comparisons for data collected by triad at MLO? The traveling Brewer reference is used to calibrate station instruments. It would be good to include its record with respect to BrT in this paper. Here are specific comments: 1) line 68. The text "230 Brewer instruments deployed" is in contradiction with the abstract where 230 instruments are referred to as "produced". Were all produced instruments deployed? 2) Line 70-71. The paper states that 123 instruments are currently in operations and are located at 88 stations. How many countries use Brewer instruments for ozone monitoring? Are there Brewers that are not part of the WMO GAW network and do not submit data to WOUDC for archiving? 3) Lines 73-74. Do I understand correctly that effective ozone cross-section is determined once after the instruments are produced? Are there in-field instrument adjustments that can change the instrument-specific absorption cross-section, overtime, or abruptly? Is there a method to check the stability of the ozone cross-sections? Is it done when the instrument is calibrated at MLO? 4) Line 80. "reference instrument . . .is independently calibrated every 2-6 years". What are the WMO GAW requirements for the frequency of calibrations of the triad? Is it consistent with the requirement for the in-field instrument calibrations? Table 1 shows that some instruments were not calibrated for 6 years. Would this affect the triad stability? What is the requirement for the

traveling standard calibration? 5) Line 109, Table 1, right column, row 6 – "Significantly less instrumental stray light than in single instrument" – please quantify what it means, include information about the level of rejection of the stray light, i.e. 10ˆ-4, 10ˆ-5 in the wings? Is stray light here attributed to the out-of-of band light? How much does it contribute to the total column ozone error at representative air mass over Toronto? 6) Line 145. The period of evaluation includes 2019 which is after the BrT was moved to a different location in 2018. Why not exclude 2018-2019? 7) Line 170 – "seasonal mean" – is it the same 3-month averages that are discussed later (line 432)? 8) Line 181, another mentioning of the "good stray-light control". Please be more specific. In Zhao et al. (2016) "good" is defined as low AMF dependence up to 81.6 degrees SZA, or within 1% up to AMF=5.5 9) Lines 197-198, "bi-weekly" means two weeks? Are you referring to the fact that the SBUV total ozone data are selected within the box centered on the station location, +/2 degrees in latitude and +/-20 degrees in longitude, and then distance weighted to create the station overpass? What is the uncertainty of SBUV total ozone overpass over Toronto? When comparing to satellite overpass data, do you use the satellite data uncertainty in the estimate of the agreement with Brewers? 10) Line 200, the reference to "+/- 1%" is one or 2 standard deviation? This number is based on the monthly averaged comparisons. How does it compare to the results in Table 5 where one standard deviation is provided based on 3-month averaged data? 11) Lines 251-232. Please explain why the instrument with more points would not dominate the forming of the baseline. Is it in reference to the previous method where three Brewers are used to establish a baseline? In the 3d party method, the baseline is derived for each instrument separately, therefore the 3d party instrument represents the "baseline"? 12) Line 268. In this method, B and C are shared between the instruments. In case one of the instruments have a stray light contribution that is larger than in the other two instruments, would it create the offset in the B and C coefficients? Is there a weighting method used to determine these coefficients? 13) Line 288. Would the effective absorption cross-section value change with the solar zenith angle due to the presence of the stray light? Do you restrict data comparisons to SZAs that have

limited impact of the stray light? 14) Line 319 – "only good quality satellite data are used in the analyses". What are these criteria? Please discuss the QA criteria (flags) in section 2.4. 15) Line 341. What was the reason to select 3-months averages for the presentation of results? 16) Line 357-358. What is the 3-month mean TO and mean air mass in Toronto in each season? Is it comparable to 330 DU and mu=2? 17) Lines 364-369. Figure 2 suggests a drift in Brewer #14 between 1999-2004 and in Brewer #8 between 2007-2013. Were the drifts corrected in the data archived in WOUDC? According to Table 2, Brewer # 14 was calibrated in 2000, 2005, 2008, and 2013. If the drift is detected between independent calibrations, is there a method to post correct the data prior to the latest calibration reference? Brewer #145 shows a large spike in both errors with the opposite sign. What caused it? 18) Line 383, "although empirical correction methods have been applied, the residual effect still exists". Figure 3 shows sudden changes in biases in 2016 and 2017, winter season. Does it have anything to do with this Brewer calibration in 2015? Can you explain instrument issues in this section while discussing Figure 3? 19) Line 412, Was eq(3) use to derive errors in the ETC and ozone absorption cross-sections? Was total ozone from Pandora used for this assessment? 20) Line 426, When issues with ozone absorption cross-section for Brewer 145 are discussed, what period if referred to? It is not clear from Figure 5(d) 21) Line 453, Table 5 results need to be discussed in greater detail. For example, if all comparison periods are included in the assessment of the BrT's errors relative to satellite overpass data, the mean bias increases to 0.625 %, which is larger than the BrT-D bias. Another interesting fact is that OMITO3 shows the largest bias from both BrT and BrT-D, whereas OMDOAS bias is much lower. TROPOMI bias is negative wrt BrT-D, and it is almost of the same magnitude as of the OMDOAO3, but of the opposite sign. Are the OMDOAO3 and TROPOMI biases related to TRPOMI higher spatial resolution or their respective ozone absorption cross-sections? There seems to be a difference in relative biases for BrT and BrT-D, where BrT-D is often higher (although the difference is not statistically significant). Is there any reason for this? It would be of interest to know of each Brewer calibration results and how much the calibration was changed.

22) Line 463, " the same long-term stability of the Brewer reference instruments when compared with Pandora or satellite instruments". Figure 7 indicates that the mean bias (by eye) of Brewers in 1999-2004 is near 0% relative to MERRA-2, then it changes to ∼2% in 2005 (MERRA-2 change?). The bias in 2005-2015 shows a slow ∼ 1% drift. There is a step-change in 2015 and then it rises to ∼1% in 2017. Brewer #15 is the lowest in 2006. Brewer #08 is the lowest in 2017-2018 Are all these differences related to the MERRA-2 changes of assimilated data? 23) Line 476, after October 2004 instead of 2014? 24) Line 505-508. The issue with strong temperature dependence in Brewer #15 is discussed. The optical frame was fixed in 2017. Was the data prior to 2017 corrected? 25) Lines 508-510. Wavelength drift in Brewer #145 is discussed. It would make sense to mention instrumental issues while discussing results in Figures 3 and 6. 26) Line 513 – Hardware replacement issues, ie. ND filter and mercury bulb. What is the recommendation to the BrT and BrT-D data reprocessing? Please make sure to refer here to Appendix B.

Data availability section: There is no link to TROPOMI data Brewer data for triad is not accessible through WOUDC.
* * *

---

## Referee Comment (RC2) · Anonymous Referee #2 · 18 Oct 2020

This is a good update of the work of Fioletov et al. [2005] addressing the precision on the WCC triad, with interesting model comparison introducing external instruments in the assessment. However, three important topics are not addressed in this work,

1. How absolute calibration is done. 2. How the calibration are maintained between absolute calibrations. 3. How the calibration is transferred to the traveling instrument and then to the Brewer network.

Simultaneous observations are required for the calibration transfer of the Brewer, so it seems feasible to have enough simultaneous measurements over a month to derive the calibration constants of the Brewer triads, using every Brewer as a reference to

calibrate the others. This will produce a monthly series of the calibrations constants (Fo, $\alpha$) to compare with model results.

There is no mention of the number of observations in the study. In contrast with other studies there is no plot of the simultaneous measurements (see for example Figure 3 of [Stübi et al., 2017]). Observing at the hourly data set used for the comparison with the reanalysis, we can almost get a view of the differences without using any average.

In general, the figures are difficult to see , especially if they are printed, because the several curves in the figure are not easily to distinguished. I suggest extending both axis for a clearer view, and using consistent symbols for BrT and BrD representation. In addition, I also suggest indicating the dates of the calibrations on the graphs.

General Comments:

1. The independent calibration of the instruments is not described. As the authors say, (line 80) The absolute calibration is "critical to review and assess the ... instrument performance", but there is no description of the methodology used, the results of the calibration and the level of agreement with the results of this work.

2. The number of calibrations of the instruments is slow, in the period of 20 years analyzed BrT instruments were calibrated four times, on average every 5 years. While brewer instruments of the Network for detection of the Atmospheric Climate Change are requested to be calibrated every year and WMO recommends a two-year cycle calibration. It is crucial to know how the calibrations are maintained between absolute calibrations.

3. The transfer method from the triad to the travelling reference need to be clarified. Which of the instruments are used for transfer ?. What ozone data do you use for the transfer? That from the BrT or the straylight-free data ? The observations from the BrT, BrT-D or an average of all six instruments? Which period of time is used for the calibration of the traveling reference.

4. Different updated versions of the model Fioletov et al. [2005] have been used to

establish the perfor- mance of the Brewer instrument, but this method is not used for the satellite and reanalysis comparison. For validation of this model a comparison of the triads using hourly observations (as reanalysis ) may be of interest.

5. The Methods 2 and 3 also evaluate the error in the Extraterrestrial constant and absorption coefficient. These parameters are also obtained during the calibration, but no comparison is made between the model-derived parameters and those obtained when the instrument is calibrated.

6. The Stray light effect on the ozone is the power law of the ozone slant column Karppinen et al. [2015] Moeini et al. [2019], although the observations are limited by air mass (3.5) and not by ozone slant column. A Brt to BrtD comparison against the ozone slant column may give us the correct limitation of the ozone slant column for the analysis.

7. The use of different timescales, monthly, three monthly or six monthly make the comparison of the different models difficult. Please unify the results.

8. Results of the regular standard lamp tests of the Brewers, normally a good indicator of the stability of the instrumental calibration. A comparison of these measured SL-test records with the presented statistical parameters should be included and hopefully show the same good stability.

Specific Comments

3.1. Page 1 Line 27 Reference to the WMO requirements document is missing.

3.2. Page 1 Line 27 Reference to the uncertainty analysis is missing.

3.3. Page 2 Line 49 random uncertainty ? Please use standard meteorologic terminology

3.4. Page 2 Line 53 Please update Stray Light correction references, [Karppinen et al., 2015], [Rimmer et al., 2018]

3.5. Page 3 Line 63 The Arosa triad is now in Davos at PMOD World Radiation Center ([Stübi et al., 2017])

3.6. Page 3 Line 65 Reference comparisons are described in [Redondas et al., 2018]

3.7. Page 3,Line 80 The instrument calibration every 2-6 years ?, the range looks 3-8 years.

3.8. Page 5,Line 116 Please detail the configuration of the BPS. Was this software also used for the previous Fioletov et al. [2005] analysis? Which are the main differences?

3.9. Page 5,Line 113 Please associate the references with the corresponding product

3.10. Page 6,Line 140 What is an independent calibration technique? Please clarify.

3.11. Page 7,Line 180 Please indicate the Pandora calibration.

3.12. Page 8,Line 165 Are Serdyuchenko cross sections used in this work? Please clarify.

3.13. Page 8,Line 170 Can you please summarize the differences between the official Pandora observations at Downsview that can be obtained from the Pandonia Global Network, and the ones used in this work? Are the observations used here also publicly available?

3.14. Page 8,Line 170 StrayLight (ozone slant column dependence) , see general note 9.

3.15. Page 10,Line 220 Can you quantify the good quality of MERRA total ozone , for example, the BIAS and standard deviation with ground base?.

3.16. Page 10,Line 245 "the baseline is only needed to adjust for the time difference in ozone measurements by individual Brewers" How large is the time difference between the measurements of the Brewers of the Triads? Couldn't they run in sync? If all the Brewers were in sync, would the baseline calculation (Ai coefficients) still be needed?

3.17. Page 12,Line 307 As the triads receive its ETC independently , can be used as ozone for the model 3.

3.18. Page 13,Line 325 The total ozone above 400 DU are usual in Toronto and with 3.5 airmass limit means 1400 ozone slant column, so this observations are seriously affected by stray ligth. Why the double brewer are also limited in airmass?.

3.19. Page 14,Line 341 $\sigma$′ is not defined, is it the mean? In that case, it would be better to use $\sigma$ ÌĎ

3.20. Page 15,Line 358 It looks like there is a factor 10 missing on the formula.

3.21. Page 15,Line 360 For the uncertainties of ETC, the goal is to have it within $\pm$ 5 R6 ratio units. Please can you clarify which are the typical conditions, and how are these threshold parameters are obtained?

3.22. Page 15,370 The goal of ETC and ozone absorption coefficient should be plotted also as reference.

3.23. Page 16,Figure 2 Could you add the calibration dates to this figure? For Brewer #008, it looks like the error is increasing over the last three years of the period between the 2008 and 2015 calibrations?

3.24. Page 17, Line 395 Figure is difficult to see.

3.25. Page 18, Line 420 Table 4: for comparison, we suggest to include the results of Model 2

3.26. Page 19, Figure 5 It is difficult to see anything, probably it would be better to have one plot for every satellite.

3.27. Page 20, Figure 6 Could you please add a plot with the standard deviation?

3.28. Page 23, Line 515 There is a correction method to account for the filter no linearity Rimmer et al. [2018] why is not applied ?

3.29. Page 24, Line 523 Please clarify, the instrument is not described before and it is not clear how affect to the ozone measurement and when this issue affects to the results.

3.30. Page 24, Line 530 See General comment 8

3.31. Page 24, Line 545 Please explain how Figure 8b is obtained.

3.32. Page 24, Line 575 The determination with the model 2 of the ETC and ozone absorption coefficients cannot be defined as error budget- The results of the Model 2 are quite far from the goal (the axis limits of Figure 2 are +/- 100 R6 ETC units but the goal is +/-5 R6 units).

3.33. Page 26, Line 585 The uncertainty of the Brewer triad is not established on this work, only its long term precision. The highly precise "group scan" is not discussed on this work and shouldn't be in the conclusions.

References

V. E. Fioletov, J. B. Kerr, C. T. McElroy, D. I. Wardle, V. Savastiouk, and T. S. Grajnar. The Brewer reference triad. Geophysical Research Letters, 32(20):L20805, October 2005. ISSN 1944-8007. https://doi.org/10.5194/amt-2020-32410.1029/2005GL024244. URL http://onlinelibrary. wiley.com/doi/10.1029/2005GL024244/abstract.

Tomi Karppinen, Alberto Redondas, Rosa D. García, Kaisa Lakkala, C. T. McElroy, and Esko Kyrö. Compen- sating for the Effects of Stray Light in Single-Monochromator Brewer Spectrophotometer Ozone Retrieval. Atmosphere-Ocean, 53(1):66–73, January 2015. ISSN 0705-5900, 1480-9214. https://doi.org/10.5194/amt- 2020-32410.1080/07055900.2013.871499. URL http://www.tandfonline.com/doi/abs/10. 1080/07055900.2013.871499.

Omid Moeini, Zahra Vaziri Zanjani, C. Thomas McElroy, David W. Tarasick, Robert D. Evans, Irina Petropavlovskikh, and Keh-Harng Feng. The effect of instrumental stray light on Brewer and Dobson total ozone measurements. Atmospheric Measurement Techniques, 12(1):327–343, January 2019. ISSN 5 1867-1381. https://doi.org/10.5194/amt-2020-324https://doi.org/10.5194/amt-12-327-2019. URL https: //amt.copernicus.org/articles/12/327/2019/. Publisher: Copernicus GmbH.

Alberto Redondas, Virgilio Carreño, Sergio F. León-Luis, Bentorey Hernández-Cruz, Javier López-Solano, Juan J. Rodriguez-Franco, José M. Vilaplana, Julian Gröbner, John Rimmer, Alkiviadis F. Bais, Vladimir Savastiouk, Juan R. Moreta, Lamine Boulkelia, Nis Jepsen, Keith M. Wilson, Vadim Shirotov, and Tomi Karpinnen. EUBREWNET RBCC-E Huelva 2015 Ozone Brewer Intercomparison. Atmospheric Chemistry and Physics, 18(13):9441–9455, July 2018. ISSN 1680-7316. https://doi.org/10.5194/amt-2020- 324https://doi.org/10.5194/acp-18-9441-2018. URL https://www.atmos-chem-phys.net/18/ 9441/2018/acp-18-9441-2018.html.

John S. Rimmer, Alberto Redondas, and Tomi Karpinnen. EuBrewNet â A European Brewer network (COST Action ES1207), an overview. Atmospheric Chemistry and Physics, 18(14):10347–10353, July 2018. ISSN 1680-7316. https://doi.org/10.5194/amt-2020-32410.5194/acp-18-10347-2018.

René Stübi, Herbert Schill, Jörg Klausen, Laurent Vuilleumier, Julian Gröbner, Luca Egli, and Dominique Ruffieux. On the compatibility of Brewer total column ozone measurements in two adjacent valleys (Arosa and Davos) in the Swiss Alps. Atmospheric Measurement Techniques, 10(11):4479–4490, November 2017. ISSN 1867-8548. https://doi.org/10.5194/amt-2020-32410.5194/amt-10-4479-2017. URL https: //amt.copernicus.org/articles/10/4479/2017/.

Please also note the supplement to this comment:
https://amt.copernicus.org/preprints/amt-2020-324/amt-2020-324-RC2-supplement.pdf
* * *

---

## Short Comment (SC1) · 5 Nov 2020

**Comments to:**

**The world Brewer reference triad – updated performance assessment and new double triad**

**Xiaoyi Zhao, Vitali Fioletov, Michael Brohart, Volodya Savastiouk, Ihab Abboud, Akira Ogyu, Jonathan Davies, Reno Sit, Sum Chi Lee, Alexander Cede, Martin Tiefengraber, Moritz Mülle, Debora Griffin, and Chris McLinden**

https://doi.org/10.5194/amt-2020-324

Sergio Fabián León-Luis[1,2], Alberto Redondas[1,2], Virgilio  Carreño[1,2], Javier López-Solano[3,2], Alberto Berjón[3,2], and Daniel Santana-Díaz[4,2]

[1]Izaña Atmospheric Research Center, Agencia Estatal de Meteorología AEMET, Tenerife , Spain
[2]Regional Brewer Calibration Center for Europe, Izaña Atmospheric Research Center, Tenerife, Spain
[3]Tragsatec, Madrid, Spain
[4]LuftBlick, Mutters, Austria

**Correspondence:** Alberto Redondas (aredondasm@aemet.es)

**1   Comparison with the RBBC-E triad in León-Luis *et al.* (2018)**

In the paper the authors claim that the comparison with the RBCC-E Triad presented in León-Luis *et al.* (2018) should not be carried out because the calculation is not consistent with the results of Model 1 in the present paper by Zhao et al. Note Model 1 was proposed in Fioletov *et al.* (2015).

In León-Luis *et al.* (2018), we calculate a quadratic polynomial fit for every Brewer as

$$O_3 = A + B \cdot (t - t_0) + C \cdot (t - t_0)^2 \tag{1}$$

obtaining for each instrument the corresponding values of $A$, $B$ and $C$. Model 1 in Fioletov *et al.* (2015) however calculates common $B$ and $C$ values for all instruments.

We take the opportunity of the open discussion of this paper to update the calculations of the RBCC-E Triad to be consistent with Fioletov *et al.* (2015), and also to compare the results of both Eq. 1 and Model 1 from Fioletov *et al.* (2015).

Table 1 contains the 3-month standard deviation of the $A_i$ coefficients obtained when the RBCC-E data are re-evaluated using Model 1, together with our previous published results. As can be observed, the values for each Brewer change slightly, depending on the method applied. However, the mean value of the Triad is similar, 0.23% versus 0.27%. This result confirms that there is very little difference between both methods when are applied to the RBCC-E Triad data.

This point can be better understood with an example. Fig. 1 demonstrates the total ozone column recorded on November 16th, 2016 (Fig. 4 in León-Luis *et al.* (2018)), where the data have been fitted used the two methods previously described. Table 2 contains the $A$, $B$ and $C$ coefficients calculated by both methods. As can be seen, regardless of the method used,

**Table 1.** RBCC-E and World Reference Triads: 3-month standard deviation. We include the values of the World Reference Triad from Zhao *et al.* (2020) for comparison.

| | RBCC-E | | World Reference | | | |
|---|---|---|---|---|---|---|
| Brewer | $\sigma_{3month}$, Eq.1 | $\sigma_{3month}$, Model 1 | Brewer | $\sigma_{3month}$, Model 1 | Brewer | $\sigma_{3month}$, Model 1 |
| #157 | 0.20 [1] | 0.19 | #008 | 0.43 (0.40) | #145 | 0.44 |
| #183 | 0.31 | 0.26 | #014 | 0.36 (0.46) | #187 | 0.26 |
| #185 | 0.29 | 0.23 | #015 | 0.42 (0.39) | #191 | 0.33 |
| Mean | 0.27 | 0.23 | | 0.40 (0.42) | | 0.34 |

Note: The standard deviations of Brewers 157 and 185 were interchanged in Table 5 of reference León-Luis *et al.* (2018)

[Figure]

**Figure 1.** Ozone values measured on November 16th, 2016, marked with circles. Solid and dotted lines correspond to the $2^{nd}$ order polynomial fitted using Eq. 1 (RBCC-E method) and Model 1 from Fioletov *et al.* (2015) (World Reference Model method), the Time units are the minutes from solar noon. The $A$ coefficients calculated with both methods are also shown.

the derived $A$ coefficients are very similar. Therefore, the mean daily value of the RBCC-E Triad, the relative errors for each instrument, and the standard deviation, calculated from these coefficients, should not differ significantly. Furthermore, the $B$ and $C$ coefficients calculated by both methods are similar, which suggests that the adjusted functions will exhibit the same behavior as shown the Fig. 1. In conclusion, although both calculation methods are not the same, the results in the case of the RBCC-E Triad are very close. A similar result is achieved when no mathematical adjustment is used and the mean from the simultaneous measurements is calculated directly.

**Table 2.** Coefficients calculated with the two methods for the RBCC-E Triad data of November 16th, 2016

| | $A$, $B$, and $C$ coefficients |
|---|---|
| RBCC-E | $A_{157} = 276.53$, $B_{157} = -0.0040$, $C_{157} = -4.855e-5$
$A_{183} = 278.72$, $B_{183} = -0.0025$, $C_{183} = -6.337e-5$
$A_{185} = 277.42$, $B_{185} = -0.0028$, $C_{185} = -6.033e-5$ |
| World Reference | $A_{157} = 276.76$, $A_{183} = 278.60$, $A_{185} = 277.35$
$B = -0.0030$, $C = -5.8122e-5$ |

**Table 3.** Percentage difference of the mean of the three instruments, mean and its standard deviation and the percentage of observations 1% 0.5% and 0.25% of the five minutes simultaneous measurements and daily mean

| | Brewer | Mean | $\sigma$ | <1% | <0.5% | <0.25% |
|---|---|---|---|---|---|---|
| 5 min | #157 | -0.041 | 0.342 | 0.994 | 0.909 | 0.687 |
| | #183 | 0.023 | 0.372 | 0.991 | 0.900 | 0.701 |
| | #185 | 0.018 | 0.342 | 0.99 | 0.921 | 0.758 |
| daily | #157 | -0.002 | 0.245 | 0.999 | 0.979 | 0.816 |
| | #183 | -0.005 | 0.309 | 0.999 | 0.931 | 0.757 |
| | #185 | 0.007 | 0.267 | 0.992 | 0.954 | 0.866 |

Table 3 shows the mean and standard deviation of the ratios for the 5 minutes simultaneous measurements and daily mean values and note that the standard deviations values in this table are fairly similar to those in Table 1, even though the periods used for the calculations are not the same (5 minutes, daily and 3 months).

Up to this point, we have shown that both methods produce a similar result. The difference between the standard deviation reported by both Brewer Triads could then be associated to others factors which have not been considered in these works, such as e.g. the intra-day ozone variability or the number of ozone (Direct Sun) measurements made per day at each station. These factors can affect the robustness of the mathematical fits and, hence, introduce small differences between the calculated $A$ coefficients that are difficult to evaluate for two stations so far apart.

**2   Additional comments**

In this section we include some other comments to Zhao *et al.* (2020), but first we want to acknowledge the effort of the World Reference Triad to maintain all these instruments during decades with such a high precision. Once the precision of the Triads has been established the challenge is to quantify the uncertainty, especially that produced by the described instrumental issues and include them in the analysis.

1.  We do not agree with the comment that the 0.5% level cannot be achieved due to limitations of the Brewer hardware. Some of the issues described, such as for example the filter non linearity, can be addressed, and indeed are accounted

for in the processing performed at Eubrewnet. Eubrewnet's processing also takes into account the issue described for Brewer #15 – the observations not compensated with mercury tests are automatically filtered out.

2. The cited Pandora manual has more than 150 pages, so it is difficult to find the ozone processing details. It could be better to refer to the ozone processing in the user guidelines avaliable at https://www.pandonia-global-network.org/wp-content/uploads/2020/01/LuftBlick_FRM4AQ_PGNUserGuidelines_RP_2019009_v1.pdf. Furthermore, if we understand it correctly, the data used in the present paper by Zhao et al. is not the operational one that is available to the public for download.

3. It looks that there is a trend on the Merra comparison from 2005 to 2015, with Brewer #015 going from +2% to -2%

4. **Appendix A**. The standard deviation of the ozone measurement is strongly affected by clouds and is also used as cloud mask to filter the AOD measurements affected by rapid moving clouds (López-Solano *et al.*, 2017). Some of the brewer are equipped with full sky cameras, are the observations reported in Zhao *et al.* (2020) also filtered by clouds ?

5. **Appendix A**. Figure A2 shows the dependence with the ozone air mass factor (AMF), as the stray-light is a function of AMF (Karppinen *et al.* (2015)) , but in the text the discussion is focused on the solar zenith angle and air mass

6. **Appendix A**. An statistical approach to estimate the single triad stray light Diemoz *et al.* (2015) or the determination of the empirical correction by comparison with the double one (Redondas *et al.*, 2018) could be performed to the dataset.

**References**

55 Diémoz, H., Redondas, A., Karppinen, T., Siani, A. M., Casale, G. R., Scarlatti, F., Eleftheratos, K., Stanek, M. and San Atanasio, J. M.: Statistics-based evaluation of the stray-light effect on ozone measurements in the framework of the European Brewer Network (EU-BREWNET), [online] Available from: https://repositorio.aemet.es/handle/20.500.11765/11816 (Accessed 3 November 2020), 2015.

Fioletov, V. E., Kerr, J., McElroy, C., Wardle, D., Savastiouk, V., and Grajnar, T.: The Brewer reference triad, Geophys. Res. Lett., 32, L20805, https://doi.org/10.1029/2005GL024244, 2005.

60 Karppinen, T., Redondas, A., García, R. D., Lakkala, K., McElroy, C. T. and Kyrö, E.: Compensating for the Effects of Stray Light in Single-Monochromator Brewer Spectrophotometer Ozone Retrieval, Atmosphere-Ocean, 53(1), 66–73, doi:10.1080/07055900.2013.871499, 2015.

León-Luis, S. F., Redondas, A., Carreño, V., López-Solano, J., Berjón, A., Hernández-Cruz, B. and Santana-Díaz, D.: Internal consistency of the Regional Brewer Calibration Centre for Europe triad during the period 2005–2016, Atmospheric Measurement Techniques, 11(7),

65 4059–4072, doi:https://doi.org/10.5194/amt-11-4059-2018, 2018.

López-Solano, J., Redondas, A., Carlund, T., Rodriguez-Franco, J. J., Diémoz, H., León-Luis, S. F., Hernández-Cruz, B., Guirado-Fuentes, C., Kouremeti, N., Gröbner, J., Kazadzis, S., Carreño, V., Berjón, A., Santana-Díaz, D., Rodríguez-Valido, M., De Bock, V., Moreta, J. R., Rimmer, J., Smedley, A. R. D., Boulkelia, L., Jepsen, N., Eriksen, P., Bais, A. F., Shirotov, V., Vilaplana, J. M., Wilson, K. M. and Karppinen, T.: Aerosol optical depth in the European Brewer Network, Atmos. Chem. Phys., 18(6), 3885–3902, doi:10.5194/acp-18-

70 3885-2018, 2018.

Redondas, A., Carreño, V., León-Luis, S. F., Hernández-Cruz, B., López-Solano, J., Rodriguez-Franco, J. J., Vilaplana, J. M., Gröbner, J., Rimmer, J., Bais, A. F., Savastiouk, V., Moreta, J. R., Boulkelia, L., Jepsen, N., Wilson, K. M., Shirotov, V. and Karppinen, T.: EUBREWNET RBCC-E Huelva 2015 Ozone Brewer Intercomparison, Atmospheric Chemistry and Physics, 18(13), 9441–9455, doi:https://doi.org/10.5194/acp-18-9441-2018, 2018.

75 Zhao, X. and Fioletov, V. and Brohart, M. and Savastiouk, V. and Abboud, I. and Ogyu, A. and Davies, J. and Sit, R. and Lee, S. C. and Cede, A. and Tiefengraber, M. and Müller, M. and Griffin, D. and McLinden, C.: The world Brewer reference triad – updated performance assessment and new double triad., Atmospheric Measurement Techniques Discussions, https://doi.org/10.5194/amt-2020-324.

---

## Author Comment (AC1) · 27 Dec 2020

**Response to Referee #1:**

We thank referee #1 for their helpful comments. Our responses are given below in black with the referee's comments in blue. The new text in the modified manuscript is given in red (italicized).

This is the first review of the paper "The world Brewer reference triad – updated performance assessment and new double triad by Xiaoyi Zhao et al. This paper is of great importance for the WMO Brewer network as it discusses the stability of the world Brewer triad maintained by the ECCC, Canada. Comparisons between the single Brewer triad (BrT) and the double Brewer triad (BrT-D) are reported for the 1999-2019 period. The previous assessment of the BrT performance (Fioletov et al., 2005) is used to verify the stability of the reference instruments over an extended period (1984-2019). Four statistical methods to evaluate the uncertainty of each instrument relative to the BrT and BrT-D baseline, to the independent reference observations (Pandora and eleven satellite records), and to the reanalyses (MERRA-2) are presented and summarized in plots and tables.

The paper is well written, the figure used to demonstrate the analyses are clear. The summary tables support the discussion and allow us to evaluate stability and random uncertainties of the total ozone observation originating from uncertainties in the extra-terrestrial constant (ETC) and the effective absorption cross-section coefficient specific to each instrument in the triad. There are a couple of inconsistencies in the analyses, including grouping of the data in either monthly, 3-months, or 6-months averages. It is not clear why the time periods for averages are changing depending on the analyses. It would make sense to present all data as monthly averages.

We thank referee #1 for the positive feedback on this work. As pointed out by the referee, some of the analyses were done at different frequencies. The analyses made with Models 1 and 2 as well as those analyses with satellite and reanalysis data used a 3-month frequency (e.g., Figs. 1a, 2, 5, 6, and 7). The 3-month frequency is selected due to having a better balance of sufficient co-incident measurements and good temporal resolution. For example, Fig. R1 shows

the number of days that can be used in Model 1 analysis with different analysis frequency (from 1 month to 6 months). A specific day is analyzed with Model 1 only if each of the three instruments has 1) at least ten measurements on that day and 2) at least three measurements in each half-day on that day (see Section 4.1.1). The median values of days used in Model 1 analysis are 11, 32, and 64 for analysis frequencies of 1 month, 3 months, and 6 months, respectively. Using monthly averages will have some undersampling issues, especially in the winter period. In addition, the Models 1 and 2 analyses done in the previous triad assessment (Fioletov et al., 2005) used a 3-month frequency, which was selected to preserve any possible artificial seasonal cycle in ETC errors and also to have as many data points as possible. Thus, to make this new assessment work be directly comparable with the first assessment, we decided to keep using this 3-month frequency, and change other analyses to match this frequency.

[Figure]

**Figure R1. The number of days included in Model 1 analysis for Brewer reference Triad (BrT).**

Different frequencies were used when comparing with Pandora data. For example, we selected monthly frequency (Fig. 3, relative difference) to better illustrate the fine-scale variability (e.g., January to February 2017, in the original Fig. 3). However, we agree with the referee that a consistent analysis frequency is a better choice. Thus, Fig. 3 has been modified to a 3-month frequency.

[Figure]

**Figure 3.** **3-month** relative differences between Brewers and Pandora total column ozone. **3-month** averages are calculated if there are at least ten coincident measurements between Brewer and Pandora for that period. The black dash line represents the time when BrT-D was relocated to Egbert, i.e., Pandora and BrT-D were not co-located.

We also updated Fig. 4 to use a 3-month frequency.

[Figure]

**Figure 4.** Relative systematic uncertainties in ETCs and effective ozone absorption coefficients estimated using Model 3. Description of y-axes is in Fig. 2. Each point on the graph represents a **3-month** average. The black dash line represents the time when BrT-D was relocated to Egbert.

The 2005 paper analyzed data starting in 1984. Why does this paper exclude the 1984-1988 period? Since the triad is independently calibrated at Mauna-Loa observatory, where the

station Dobson (since 1957) is located, why not to perform comparisons for data collected by triad at MLO? The traveling Brewer reference is used to calibrate station instruments. It would be good to include its record with respect to BrT in this paper.

The suggestions from the referee, i.e., including 1984-1998 data and comparisons with MLO data are very important. We think that the whole four decades of observations should be carefully evaluated and can be useful, e.g., to provide high-precision TCO trends in Toronto. However, the focus of the current work is to provide an updated assessment for triad in the past two decades. Thus, we selected the 1999-2019 period in this assessment work to provide 5 years of overlap with the first assessment (1984-2004).

A comparison between Brewer reference instruments and Dobson instruments at MLO is also possible. However, for each calibration trip, the Brewer reference instrument will only co-locate with Dobson at MLO for about a month. Thus, the dataset will be small, i.e., less than 17 months (see Table 2). Including these analyses will not likely affect the results and conclusion from this work. Moreover, the Dobson operated at MLO is not the Dobson world reference instrument. The world reference, Dobson #83, is calibrated at MLO once every several years. Therefore, it is not possible to compare the triad instruments with the world reference Dobson. Thus, to make the current work more concise, we would prefer to leave this analysis work in a future publication.

In addition, in a joint work with other Brewer groups, we are planning a publication detailing about the absolute calibration procedure, the calibration transfer procedure, and an assessment of travelling standard instruments soon. Together with the triad assessments (Fioletov et al., 2005 and current work), these works will provide the general, but important pictures, of ozone monitoring activities carried out by the global Brewer network.

Here are specific comments:

1) line 68. The text "230 Brewer instruments deployed" is in contradiction with the abstract where 230 instruments are referred to as "produced". Were all produced instruments deployed?

Some of the Brewers have been retired after years of services and are not currently deployed. The sentence has been modified.

*By 2019, there were more than 230 Brewer instruments manufactured, with most of them deployed worldwide within the WMO GAW global ozone monitoring network.*

2) Line 70-71. The paper states that 123 instruments are currently in operations and are located at 88 stations. How many countries use Brewer instruments for ozone monitoring? Are there Brewers that are not part of the WMO GAW network and do not submit data to WOUDC for archiving?

This is an interesting question, but outside the scope of this paper. Some instruments are operated by universities and have no connections to the WMO GAW. We can only provide information about sites that were calibrated using the Toronto Brewer triad as a reference. Detailed information on such calibrations and data submissions to the WOUDC is available from International Ozone Services Inc. (IOS) web site at https://www.io3.ca/Calibrations. In the last twenty years, the total number of distinct Brewers that have been calibrated by IOS is 148. On average, IOS has transferred world reference instruments' calibration to about 40 Brewers by visiting 15 countries per year. These Brewers are located in 48 countries. To complete all these calibration, IOS took 599 trips. Figure R2 shows the time series of these calibration activities. Some of these information have been included in the revised manuscript.

*In practice, each field Brewer instrument receives its ETC constant by comparing ozone values with those of the travelling standard instrument. The travelling standard itself is calibrated against the set of world reference instruments (i.e., world Brewer reference triad). The world reference triad data are used to calibrate the traveling standard, and the traveling is used to calibrate 30-40 Brewers per year, on average, around the world.*

[Figure]

Figure R2. Time series of the calibration transfers done by IOS from 1988 to 2020.

3) Lines 73-74. Do I understand correctly that effective ozone cross-section is determined once after the instruments are produced? Are there in-field instrument adjustments that can change the instrument-specific absorption cross-section, overtime, or abruptly? Is there a method to check the stability of the ozone cross-sections? Is it done when the instrument is calibrated at MLO?

Brewer uses BP (Bass-Paur) ozone cross-section (at 228.3° K, Bass and Paur, 1985), which was measured in the laboratory. The effective ozone cross-section mentioned by the referee should be the effective ozone absorption coefficient ($\Delta\alpha$). This coefficient is generated for each Brewer by performing the dispersion test (DSP) (Savastiouk, 2006) with the use of a group of discharge lamps (e.g. Hg, Cd, In). In general, the slit functions of the Brewer are determined by DSP. Then, $\Delta\alpha$ is calculated as the convolution of slit functions and literature ozone cross-section at the operating wavelengths. It is correct that the in-field adjustments may change $\Delta\alpha$. Thus, after each adjustment work, new $\Delta\alpha$ will be measured via DSP. This work can be done in the field. The stability of $\Delta\alpha$ is directly related to the stability of the wavelengths setting in the Brewers. This is regularly checked using the stable solar spectrum as the reference using the so-called Sun Scan test. Some of these information has been included in the revised manuscript.

*For example, the effective ozone absorption coefficients (Δα) are determined for each individual instrument in laboratories via dispersion test, and are regularly checked using the stable solar spectrum as the reference using the so-called Sun Scan test (Savastiouk, 2006).*

4) Line 80. "reference instrument … is independently calibrated every 2-6 years". What are the WMO GAW requirements for the frequency of calibrations of the triad? Is it consistent with the requirement for the infield instrument calibrations? Table 1 shows that some instruments were not calibrated for 6 years. Would this affect the triad stability? What is the requirement for the traveling standard calibration?

When the primary calibration has been done for one of the reference instruments at MLO, this instrument can be used to validate the status of other reference instruments in Toronto. Therefore, to satisfy the 2-3 year interval between calibrations requirement, it is sufficient if at least one triad Brewer is calibrated at MLO every 2-3 years.

The traveling standards need to be calibrated against a World Brewer Reference – traceable instrument before and after every calibration trip. This ensures the quality of the transferred calibration and a complete understanding of the traveling standards' performance.

5) Line 109, Table 1, right column, row 6 – "Significantly less instrumental stray light than in single instrument" – please quantify what it means, include information about the level of rejection of the stray light, i.e. 10ˆ-4, 10ˆ-5 in the wings? Is stray light here attributed to the out-of-of band light? How much does it contribute to the total column ozone error at representative air mass over Toronto?

The strength of stray light effect depends on the slant ozone amount and not on air mass. The median air mass factor over Toronto is 2 ($\mu$= 2), for which the stray light effect is weak. As illustrated in Fig. R3, BrT and BrT-D start to have more than 1% relative difference when $\mu$ > 3.5 (equivalent to slant ozone 1200 DU). Thus, data only with $\mu$ ≤ 3.5 are used in this assessment work (except Fig. A2). In conditions with representative air mass values (e.g., $\mu$ values about 2), Brewers have a median standard deviation of about 1.2 DU (see Fig. A1). Details of the stray

light issue are provided in Appendix A. Following suggestions from the referee, the description in Table 1 has been updated.

[Figure]

**Figure R3. The relative difference between BrT and BrT-D, in terms of air mass factor ($\mu$) and slant column ozone. The error bars represent 1σ of the relative difference values. The black dash lines show the -1 % relative difference.**

*Significantly less instrumental stray light (out-of-band, stray light fraction $10^{-7}$) than in the single monochromators ($10^{-5}$) (Fioletov et al., 2000).*

6) Line 145. The period of evaluation includes 2019 which is after the BrT was moved to a different location in 2018. Why not exclude 2018-2019?

It was the BrT-D that been moved to the Egbert site temporarily since September 2018. We decided to include this period to demonstrate some fine-scale (spatial) variability of stratospheric ozone field (i.e., the two monitoring sites are only 55 km apart). A detailed example is provided in Section 5 (see Fig. 8), which shows the fine-scale variation may have a significant impact on the validations of high-resolution satellite TCO product (e.g., TROPOMI).

Also, please note that for Models 1 and 2 analyses, since the baseline ozone was formed by each triad, this re-location will not affect the assessment results. Although we see the difference between BrT and BrT-D in 2019 January to February as in Figs. 3 and 4, these differences will not be reflected in Figs. 1 and 2. Thus, including this period will not affect our major conclusions about BrT and BrT-D's long-term stability (via Model 1 and 2).

7) Line 170 – "seasonal mean" – is it the same 3-month averages that are discussed later (line 432)?

No. The values were estimated with monthly averages in Zhao et al., 2016. The sentence has been modified to clarify this.

*In general, after correction, the multiplicative bias in Pandora ozone data can be decreased from 2.92 to -0.04 %, with the seasonal difference decreased from ±1.02 to ±0.25 % (see Fig. 11 in Zhao et al., 2016; i.e., comparing to Brewer, corrected Pandora data has -0.04 + 0.25% offset in summer and -0.04 – 0.25% offset in winter).*

8) Line181, another mentioning of the "good stray-light control". Please be more specific In Zhao et al. (2016) "good" is define as low AMF dependence up to 81.6 degrees SZA, or within 1% up to AMF=5.5

Done.

*The Pandora and BrT-D instruments have good stray-light control, and under typical ozone conditions (i.e., slant column ozone less than 1500 DU), their air mass dependence is comparably low up to 81.6° SZA (within 1% up to AMF = 5.5; Zhao et al., 2016).*

9) Lines 197-198, "bi-weekly" means two weeks? Are you referring to the fact that the SBUV total ozone data are selected within the box centered on the station location, +/2 degrees in latitude and +/-20 degrees in longitude, and then distance weighted to create the station overpass? What is the uncertainty of SBUV total ozone overpass over Toronto? When comparing to satellite overpass data, do you use the satellite data uncertainty in the estimate of the agreement with Brewers?

Yes, due to a small field-of-view, the SBUV instruments provide global coverage about every two weeks. In other words, for some sites, the true sampling frequency of SBUV instruments can be as low as every two weeks. Thus, the overpass algorithm is used to increase this sampling frequency to daily (Labow et al., 2013), even if the SBUV measurements were not directly overhead of the ground site. It is correct that these daily values are obtained by weighted-interpolating data measured within the box centred on the station locations. Labow et al. (2013) reported that the smoothing errors (the largest error) for total ozone retrievals are mostly less than 0.5%. The uncertainty of individual SBUV total ozone overpasses over Toronto are not available. When comparing to satellite overpass data, we did not include satellite data uncertainty in the estimate of the agreement with Brewers. Most of the published satellite data products used here (except TROPOMI) do not have reported uncertainties associated with each measurement.

*Unlike TOMS, OMI or TROPOMI, which provides daily global coverage, the non-scanning, nadir viewing SBUV instruments provide full global coverage approximately bi-weekly. The SBUV ozone column data used in this work is produced by the overpass algorithm to create daily overpass values (Labow et al., 2013;  by weighted-interpolating data measured within the box centred on the station location (±2° in latitude and ±20° degrees in longitude)).*

10) Line 200, the reference to "+/- 1%" is one or 2 standard deviation? This number is based on the monthly averaged comparisons. How does it compare to the results in Table 5 where one standard deviation is provided based on 3-month averaged data?

The ±1 % agreement reported by Labow et al. (2013) is the yearly relative difference (time series comparison) between ground-based instruments and SBUV (see their Fig. 1). There are no 3-month or 1-month standard deviations of relative differences that can be used to compare with the current study (Table 5). However, the results from Labow et al. (2013) can be compared with Fig. 5 in this study, but should be interpreted with some level of cautions. For example, the relative difference defined in this work is

$$\Delta_{rel} = \frac{Brewer - SBUV}{\frac{1}{2}(Brewer + SBUV)}.$$

Whereas in Labow et al. (2013), the relative difference was defined as:

$$\Delta_{rel} = \frac{SBUV - Brewer}{SBUV}.$$

In addition, the results in Labow et al. (2013) used an average of 33 northern hemisphere sites. Fig. 1b in Labow et al. (2013) (note 1b is the TOMS V8 total ozone data products that used in this study, 1a is the profile integrated total ozone) shows that the relative differences are in a range of -2 to 4 % (monthly mean) in 1999 to 2010 period (with yearly averages in a range of 0 to 2.5 %). The results from Fig. 5 of this work shows the 3-month relative differences are in a range of -3 to 6 % (also see Fig. R4, which only shows the results from SBUV). We also calculated yearly relative difference which shows Brewers and satellites TCO agrees well within -2 to 3 % (except for SBUV 19 in 2019, which has very sparse coincident observations), as shown in Fig. R5. Thus, we think that the comparison results in this work is in good agreement with previous studies. The description for SBUV series and Fig. 5 (see Section 4.2) has been updated.

*Labow et al. (2013) reported that the total column ozone data from Brewers and SBUVs show an agreement within ± 1 % over 40 years (1970-2010; yearly relative difference).*

*Figure 5 shows the relative differences between satellite and Brewer measurements for seasonal (3 months) values are within ±4 % and yearly values are within ±3 % (not shown here) in these two decades (1999-2019).*

[Figure]

**Figure R4. The relative difference between satellites and the world Brewer reference triads (BrT and BrT-D). Same as Fig. 5, but only shows SBUV satellites.**

[Figure]

**Figure R5. The yearly relative difference between satellites and the world Brewer reference triads (BrT and BrT-D).**

11) Lines 251-232. Please explain why the instrument with more points would not dominate the forming of the baseline. Is it in reference to the previous method where three Brewers are used to establish a baseline? In the 3d party method, the baseline is derived for each instrument separately, therefore the 3d party instrument represents the "baseline"?

The referee's comments are correct. For the Model 1 analysis, the baseline ozone is formed by fitting a $2^{nd}$ order polynomial function with observations from three Brewers for each day. Thus, if one of the three instruments produced more observations than the other two, the calculated baseline ozone will be more representative of that particular instruments (no matter if the real data quality from that instrument is good or not). This issue has been addressed by introducing the index matrix in Eqn. 2, which calculates three "baseline ozone". These three "baseline ozone" share the common curvature (i.e., $2^{nd}$ and $3^{rd}$ order terms) but have unique offsets (i.e., $A_1$, $A_2$, and $A_3$). However, it is still likely the instrument with more observations may contribute more to the curvature terms.

On the other hand, when using the third-party scheme, i.e., use Pandora TCO as the baseline ozone, we can avoid the issues mentioned above. In other words, the Brewer instrument (no matter if it has more or fewer observations in that particular day) can be "fairly" compared with baseline ozone that is independent of its own observations. The sentence has been modified as to clarify this.

*Moreover, when using* coincident *Pandora ozone data, the baseline will not have the sampling or weighting issues; i.e., the Brewer instrument that reported more data points will not dominate the forming of the baseline (i.e., as the baseline formation in Model 1, see Eqn. 2).*

12) Line 268. In this method, B and C are shared between the instruments. In case one of the instruments have a stray light contribution that is larger than in the other two instruments, would it create the offset in the B and C coefficients? Is there a weighting method used to determine these coefficients?

The referee is correct that if one of the instruments has a strong stray light issue, then it may artificially contribute to the curvature of the fitted baseline ozone. For this reason, we do not recommend to use Model 1 to analyze data measured in large AMF conditions ($\mu > 3.5$) for single Brewers. As discussed in the previous question and Appendix A, for moderate AMF ($\mu < 3.5$), both single and double Brewers have reasonable good stray light control, thus currently, we did not use any weighting method in the determination of these coefficients.

13) Line 288. Would the effective absorption cross-section value change with the solar zenith angle due to the presence of the stray light? Do you restrict data comparisons to SZAs that have limited impact of the stray light?

The effective ozone absorption coefficient ($\Delta\alpha$) is quantified by instrument slit functions (determined in DSP test) and the published ozone cross-section. The measured slit functions were acquired with discharge lamps, which might not fully represent the true slit functions of the instruments, especially when stray light became an issue. Thus, the referee is correct that $\Delta\alpha$ is different at different SZA due to stray light. To avoid this, only observations with AMF < 3.5 were included in this work to minimize chance of high slant ozone. Similar to the answer to the previous comment from the referee, we do not recommend to use measurements with large AMF in Model 2 analysis. The sentence has been modified to include these recommendations.

*Thus, the difference of total column ozone between the individual instrument and Model 1 is allocated to the "error" of ETC and effective ozone absorption values. As the stray light issue in high μ conditions may affect the formation of the baseline ozone (see Eqns. 2 and 3), all Brewer DS ozone data used in this study have μ ≤ 3.5.*

14) Line 319 – "only good quality satellite data are used in the analyses". What are these criteria? Please discuss the QA criteria (flags in section 2.4.

Done.

*The EP/TOMS total ozone data from 1996 to 2005 with a quality flag of zero were used in this work (McPeters et al., 1998).*

*The SBUV ozone column data used in this work is produced and quality assured by the overpass algorithm to create daily overpass values (Labow et al., 2013; by weighted-interpolating data measured within the box centred on the station location (±2° in latitude and ±20° degrees in longitude)).*

*In this work, OMPS-NPP L2 Nadir Mapper (NM) Ozone Total Column swath orbital v2.1 data (only good sample, with a QualityFlags of zero) from the OMPS-NM module is used.*

*In this work, the OMDOAO3 and OMTO3 OVP data are used, with L2 quality flag equal to 0 or 1 and bit 6 is not set are included (see https://avdc.gsfc.nasa.gov/pub/data/satellite/Aura/OMI/V03/L2OVP/OMDOAO3/).*

*The offline (OFFL v010107) total ozone column data (Garane et al., 2019) are used in this work (only L2 data with qa ≥ 0.75 are included).*

15) Line 341. What was the reason to select 3-months averages for the presentation of results?

This question has been addressed at the beginning of this document. A sentence has been included here.

*Using the analytical method from the first assessment work (Fioletov et al., 2005), the deviations and residuals are reported with frequencies of 3 months and 1 year, respectively, in Fig. 1. These frequencies were used because they provide a good balance between sampling frequency and sufficient co-incident measurements as well as preserve a potential seasonal component in the differences.*

16) Line 357-358. What is the 3-month mean TO and mean air mass in Toronto in each season? Is it comparable to 330 DU and mu=2?

These values (TCO = 330 DU and $\mu$ = 2) were selected based on the statistic of TCO and $\mu$ values in Toronto. The time series of 3-month TCO and mean air mass factor ($\mu$) in Toronto are shown in Fig. R6. Also, the histograms of TCO and air mass factors are shown in the right column of Fig. 6. The histograms show that 330 DU is the median TCO values in Toronto, and $\mu$ = 2 represents the mid-point of TCO seasonal (3-month) air mass variations from 1.5 to 2.5.

[Figure]

**Figure R6. Time series and histogram of TCO and air mass factor (μ) in Toronto.**

17) Lines 364-369. Figure 2 suggests a drift in Brewer #14 between 1999-2004 and in Brewer #8 between 2007-2013. Were the drifts corrected in the data archived in WOUDC? According to Table 2, Brewer # 14 was calibrated in 2000, 2005, 2008, and 2013. If the drift is detected between independent calibrations, is there a method to post correct the data prior to the latest calibration reference? Brewer #145 shows a large spike in both errors with the opposite sign. What caused it?

These drifts described by the referee (observed in Fig. 2) have not been modified since they are still within the acceptable error budgets of Brewers. As pointed out in Section 4.1.2, the errors in ETCs and ozone absorption coefficients can largely compensate for each other, thus the derived TCOs may still have "reasonable" values (i.e., 3-month deviations within ±1 %, as shown in Fig. 1). However, if the errors in ETCs and ozone absorption coefficients are too large, the TCOs measured in a day will have artificial curvature in high μ conditions. Thus, a reprocessing of data from reference instruments was made only when these errors account for more than 3 or 4 % of TCO. An example of this practice was provided in Appendix B, using Model 3. In that

case, Brewer #145's ETC error in early 2014 was as large as 4 % and the ozone absorption error was about 3 % in the operational processing version. Detailed data scrutiny was made and the cause was found (see Appendix B for more details). The re-processing was made to address the issue, with the ETC and ozone absorption errors both being decreased to the acceptable level (±2 %).

If a drift is detected between the independent calibrations, a detailed investigation will be made together with Brewer technicians and researchers. As an example provided in Appendix B, a post-correction can be made if solid evidence was not only found by models, but also confirmed by Brewer technicians.

The opposite signs in the estimated ETC and ozone absorption errors are due to the nature of the models (2 and 3). The models distribute the residuals (mismatch between observed ozone and baseline ozone) into two parts, i.e., X and Y terms in Eqns. 3 and 4. Thus, retrieved X and Y are negatively correlated. The section has been updated to reflect this information.

 *The large errors in ETCs and ozone absorption coefficients may largely compensate for each other and not be evident in the Model 1 analysis.* *This is because Model 2 distributes the residuals (mismatch between observed ozone and baseline ozone) into two parts, i.e., X and Y terms in Eqn. 3, which made the retrieved errors negatively correlated.*

18) Line 383, "although empirical correction methods have been applied, the residual effect still exists". Figure 3 shows sudden changes in biases in 2016 and 2017, winter season. Does it have anything to do with this Brewer calibration in 2015? Can you explain instrument issues in this section while discussing Figure 3?

The sudden changes in late 2016 to early 2017 were due to the "imperfect" empirical correction made for Pandora TCO (not due to Brewers). Since the effective temperature used in the Pandora TCO correction model (see Eqn. 10 in Zhao et al. 2016) is calculated with modelled ozone and temperature profiles (ERA-Interim), we might have a large bias in some extreme conditions, especially in winter (e.g., see Fig. 10 in Zhao et al. 2016). On the other hand, after correction, although Pandora TCO has reduced its seasonal deviations from Brewers, one still

can see small residuals (i.e., about 0.4 % seasonable variations, see Fig. 13c in Zhao et al. 2016). This "winter shift" is correlated to the temperatures and still can be seen in Fig. 3 (its depends on the weather; for some years, it is relatively mild).

The original Fig. 3 was made using a monthly frequency to illustrate this issue. In the new Fig. 3 (3-month, see the figure provided in the general comments part), this effect is no longer prominent. Since this feature is not prominent in the modified Fig. 3, we remove the sentence to avoid any misunderstanding of this point.

*For example, the absolute differences from the six Brewer instruments all shifted towards positive in the January to February 2017 period.*

19) Line 412, Was eq(3) use to derive errors in the ETC and ozone absorption cross-sections? Was total ozone from Pandora used for this assessment?

It was done with Eqn. 4, in which baseline ozone values are not derived from Model 1 (see Eqns. 2 and 3), but observations from Pandora. Total ozone from Pandora is used as the third-party baseline ozone in this analysis.

20) Line 426, When issues with ozone absorption cross-section for Brewer 145 are discussed, what period if referred to? It is not clear from Figure 5(d)

This was discussed with details and provided in Appendix B. Results in Fig. 5d are using the re-processed data from Brewer #145. Thus, it is correct that no clear deviations can be seen. The sentence has been modified.

*For example, when analyzing Brewer #145 data, it was revealed by the Model 3 analysis that its absorption coefficients were not ideal (in 2014, see Appendix B for more details).*

21) Line 453, Table 5 results need to be discussed in greater detail. For example, if all comparison periods are included in the assessment of the BrT's errors relative to satellite

overpass data, the mean bias increases to 0.625 %, which is larger than the BrT-D bias. Another interesting fact is that OMITO3 shows the largest bias from both BrT and BrT-D, whereas OMDOAS bias is much lower. TROPOMI bias is negative wrt BrT-D, and it is almost of the same magnitude as of the OMDOAO3, but of the opposite sign. Are the OMDOAO3 and TROPOMI biases related to TRPOMI higher spatial resolution or their respective ozone absorption cross-sections? There seems to be a difference in relative biases for BrT and BrT-D, where BrT-D is often higher (although the difference is not statistically significant) Is there any reason for this? It would be of interest to know of each Brewer calibration results and how much the calibration was changed.

It is correct that if we include results from SBUV11, SBUV14, and TOMS into the calculation, the mean bias will increase. The difference between OMDOAO3 and OMTO3 was also reported by other research works. For example, Antón et al., 2009 reported that TCO from OMTO3 is on average 2.0 % lower than Brewer data, whereas for OMDOAO3 data the bias is only 1.4 %. Thus, we think the findings here are in good agreement with previous works, i.e., OMDOAO3 (0.84 %) has a lesser bias to Brewer data than OMTO3 (1.14 %).

The comparison for TROPOMI is a bit more complicated since the viewing geometry (line of sight) plays a more important factor for such high-resolution satellite data. In this part of the work, to make it a fair comparison, we only used TROPOMI true overpass data (i.e., same as other satellites), without taking into account the difference of viewing geometries between Brewers and satellite. However, this could cause some issues when the stratospheric ozone field has a large gradient, as discussed in Section 5 (see Fig. 8). In short, the opposite signs for BrT and BrT-D's relative biases to TROPOMI should be interpreted with extra caution. In addition, to truly validate the high-resolution satellite ozone data, one will need an improved coincident data selection algorithm. We think that this result will not affect the assessment that we made for the Brewer reference instruments, which is the main goal of this project. We also decided to leave this part within the general satellite comparison and discussion sections to bring the attention of the research community to this high-resolution satellite validation topic.

Regarding the relative differences between BrT and BrT-D, we think it is better to see this effect from Figs. 3 and 7. On average, the relative differences between these two reference groups are within 1 %. Since limited by the random uncertainties of the current Brewer 5-wavelength algorithm (about 0.5 %, as discussed in Section 4.1.1), decreasing these differences will be very challenging. Some of this information has been included in the revised manuscript.

*The standard deviation of the Brewer-OMTO3 (OMDOAO3) difference (for 3-month averages) calculated for six instruments is 0.99 % (1.06 %), about 0.5 % higher than Brewers' standard random uncertainties calculated in Section 4.1.1. It is also found that Brewers have lower relative differences compared with OMDOAO3 than OMTO3, which is in agreement with previous researches(e.g., Antón et al., 2009). For high-resolution satellites, such as TROPOMI, the interpretation of the results should be made with extra cautions as the line-of-sight of ground-based and satellite instruments should be accounted for (see more details in Section 6).*

22) Line 463, " the same long-term stability of the Brewer reference instruments when compared with Pandora or satellite instruments". Figure 7 indicates that the mean bias(by eye) of Brewers in 1999-2004 is near 0% relative to MERRA-2, then it changes to ~2% in 2005 (MERRA-2 change?). The bias in 2005 2015 shows a slow ~1% drift. There is a step-change in 2015 and then it rises to 1% in 2017. Brewer #15 is the lowest in 2006. Brewer #08 is the lowest in 2017-2018 Are all these differences related to the MERRA-2 changes of assimilated data?

The 2% jump in 2005 was due to MERRA-2 changing its assimilation sources from SBUV to MLS/OMI. This information was included in the manuscript (Line 472). As shown in Table 5, we expect that the bias between SBUV and OMI will be propagated to the reanalysis data. The caption of Fig. 7 has been modified to make this more clear.

*Figure 7. The relative difference between the reference Brewers and MERRA-2 reanalysis. Each point represents a 3-month average. The green dash line represents the time when MERRA-2 changed its assimilation sources from SBUV-2 to MLS/OMI (causing about 2% relative difference). The black dash line represents the time when BrT-D was relocated to Egbert.*

There might be a slow 1% drift as described by the referee in the 2005 to 2015 period. However, given the fact that Brewers also have 0.5 % random uncertainties, we are not sure if this drift is statistically significant. Without uncertainties from the model (in addition to the propagated uncertainties from its assimilation sources), it is difficult to give any solid conclusion on such a small level of variations (i.e., if this 1 % drift is statistically significant or not). It is not always possible to determine the origin of small ±1%, differences between different datasets (for example, as in 2015-2017).

The other difference, such as Brewer #015 was the lowest in 2006, is possibly due to a real bias in the instrument. Some of these features can also be found when comparing with Pandora (see Fig. 3). However, the variations that we see in these 3-month relative difference plots are a combination of random uncertainties and biases from both Brewers and the other TCO dataset (e.g., MERRA-2 or Pandora). Similar to our answer to the previous question about satellite comparison, limited by the uncertainties in the dataset (not just from Brewers, but also from other instruments or models), detecting any variation or trend within 1% is very challenging. For example, if we simply assume that both Brewer and the other instrument have 0.7 % total uncertainty, the propagated uncertainties in relative difference will be on an order of 1%. Thus, further fine-tuning or interpretation of the current relative biases found between Brewers and other instruments (or reanalysis) may not be possible.

*The relative difference in time series is shown in Fig. 7, which demonstrated the similar long-term stability (i.e., the relative difference within ±2 %) of the Brewer reference instruments when compared with Pandora or satellite instruments.*

23) Line 476, after October 2004 instead of 2014?

Thanks. The typo has been corrected.

*For example, the mean Brewer #014 – MERRA-2 relative bias was 0.11 % ($\Delta_{rel}'$) for the SBUV-based data assimilation, but it increased to 1.07 % after October 2004, probably due to some bias in OMI data as mentioned previously in Section 4.2.*

24) Line 505-508. The issue with strong temperature dependence in Brewer #15 is discussed. The optical frame was fixed in 2017. Was the data prior to 2017 corrected?

The strong temperature dependence (TD) in Brewer #015 is not ideal, but in this case only, it had limited effects on the data. This is because the wavelength calibration tests (HG) are done regularly, which can largely reduce the impact. However, we should point out that if the time interval between the HG tests is large enough, some measurements can be affected. We included this to illustrate how Brewer hardware problems can affect the overall instrument performance. The relevant text has been modified to clarify this issue.

*For example, it was found that Brewer #015 has a particularly strong temperature dependence where the optical frame was expanding significantly faster than any other Brewer instrument. As a result, the wavelength calibration tests (HG) had to be scheduled more frequently to reduce the impact. However, we should point out that if the time interval between the HG tests is large enough, some measurements can be affected. This issue was fixed in 2017 by replacing the optical frame (details of instrument repair and upgrade history is provided in the supplementary information).*

25) Lines 508-510. Wavelength drift in Brewer #145 is discussed. It would make sense to mention instrumental issues while discussing results in Figures 3 and 6.

We agree with the referee that it makes sense to provide more details of instrumental issues while discussing the figures provided. So, we included some discussions for Brewer #015 and #145.

*A second example is the original configuration of Brewer #145 micrometer was found to have developed wear and became unreliable, causing some wavelength drifts, and as a result,*

*relatively high uncertainties for Brewer #145 as shown in Table 7 (also see larger variations of 3-month deviations from Brewer #145 compared to Brewers #187 and #191 in Fig. 1a).*

26) Line 513 – Hardware replacement issues, ie. ND filter and mercury bulb. What is the recommendation to the BrT and BrT-D data reprocessing? Please make sure to refer here to Appendix B.

We think that it is more appropriate to include some recommendations for BrT and BrT-D data reprocessing in the next paragraph.

*However, this approach raises the question of reproducibility of the obtained results and must be carefully documented. For BrT and BrT-D's data reprocessing, we recommend using the statistical models developed in relevant studies to help the identifications of potential hardware or software issues. To keep the integrity of the world reference instruments, data reprocessing could be done only if solid evidence of imperfection of hardware or software have been found and confirmed by Brewer technicians and researchers.*

Data availability section: There is no link to TROPOMI data Brewer data for triad is not accessible through WOUDC.

The Brewer triad data has been uploaded to WOUDC.

*TROPOMI data are available from http://www.tropomi.eu/data-products/total-ozone-column, last accessed: October 2020.*

---

## Author Comment (AC2) · 27 Dec 2020

**Response to Referee #2:**

We thank referee #2 for their helpful comments. Our responses are given below in black with the referee's comments in blue. The new/revised text in the modified manuscript is given in red (italicized).

This is a good update of the work of Fioletov et al. [2005] addressing the precision on the WCC triad, with interesting model comparison introducing external instruments in the assessment. However, three important topics are not addressed in this work, 1. How absolute calibration is done. 2. How the calibration are maintained between absolute calibrations. 3. How the calibration is transferred to the traveling instrument and then to the Brewer network.

As recommended by the referee, absolute calibration procedure, maintenance, calibration transfer, and assessment of travelling standard should be detailed described and published. Together with the assessment of triads, these works will provide some general, but important pictures of the Brewer ozone monitoring network. Thus, the suggested work has been included in our project plan. We will coordinate with other relevant institutes and prepare the second publication in the near future since it was discussed and recommended at the recent meeting of the WMO GAW Scientific Advisory Group on Ozone and UV. However, the purpose of this study is to demonstrate the long-term stability of the existing Brewer reference standard (the Brewer triad). Some of these information has been included in the revised manuscript.

*Thus, it is critical to review and assess the world reference instruments' performance on a regular basis. This study's focus is on the demonstration of the long-term stability of the existing reference instrument. Absolute calibration procedure, maintenance, calibration transfer, and assessment of travelling standard will be a subject of a separate study.*

Simultaneous observations are required for the calibration transfer of the Brewer, so it seems feasible to have enough simultaneous measurements over a month to derive the calibration constants of the Brewer triads, using every Brewer as a reference to calibrate the others. This will produce a monthly series of the calibrations constants ($F_0$, $\alpha$) to compare with model results.

This was essentially done by the statistical Model 2. Instead of comparing constants instrument by instrument (that makes it difficult to interpret the results), the Model 2 estimates deviations of the

constants for each instrument from the "best" value based on all measurements for each 3-month period for the entire 20-year long triad record. If a calibration constant is different from the value "prescribed" by the two other instruments, that the estimated errors for the instrument would appear as an outlier. This gives information about long-term changes in the constants and the overall triad stability.

There is no mention of the number of observations in the study. In contrast with other studies there is no plot of the simultaneous measurements (see for example Figure 3 of [Stübi et al., 2017]). Observing at the hourly data set used for the comparison with the reanalysis, we can almost get a view of the differences without using any average. In general, the figure are difficult to see, especially if they are printed, because the several curves in the figure are not easily to distinguished. I suggest extending both axis for a clearer view, and using consistent symbols for BrT and BrD representation. In addition, I also suggest indicating the dates of the calibrations on the graphs.

The analysis was done based on individual measurements and only the results are presented in the form of long-term plots. Following suggestions from the referee, we also plotted the measurements from six reference instruments with the absolute calibration dates indicated. This figure has been included in the supplementary information. In the manuscript, all six reference instruments were plotted with consistent unique colours (e.g., consistently using blue colour for Brewer #008 and consistently using red colour for Brewer #014 when results from all six instruments were presented together).

[Figure]

*Figure S1. Time series of Brewer TCO observations in Toronto. Vertical black dash lines indicate the time of primary calibrations as shown in Table 2.*

General Comments:

1. The independent calibration of the instruments is not described. As the authors say, (line 80) The absolute calibration is "critical to review and assess the ... instrument performance", but there is no description of the methodology used, the results of the calibration and the level of agreement with the results of this work.

The purpose of this work is to evaluate the triad performance based on the existing calibration results. The sentence has been modified to provide a reference to the procedures of independent calibration. The results of the independent calibration are ETC values, which have been used to produce the TCO values reported. The TCO values have been examined by Model 1 and compared with Pandora, satellites, and reanalysis data. The ETC values themselves were been evaluated via Models 2 and 3 analyses, which provided the estimated errors of the ETCs. In general, we think this work has already provided an assessment of the results of the independent calibration via the analyses mentioned above. Please note that Models 2 and 3 were designed to estimate the errors of ETC and effective ozone absorption coefficient, but not ETC or effective ozone absorption coefficient themselves.

*The extraterrestrial calibration constant (ETC) has to be determined in the field by one of the two means: 1) the independent calibration method, i.e., the Langley plot calibration method or the so-called zero airmass extrapolation technique, or 2) the calibration transfer method (e.g., transfer ETC from well-calibrated reference instruments to field instruments) (see more details about calibration procedures in Kerr, 2010).*

2. The number of calibrations of the instruments is low, in the period of 20 years analyzed BrT instruments were calibrated four times, on average every 5 years. While brewer instruments of the Network for detection of the Atmospheric Climate Change are requested to be calibrated every year and WMO recommends a two-year cycle calibration. It is crucial to know how the calibrations are maintained between absolute calibrations.

In a perfect world, Brewer would be calibrated just once and then most of the changes in the instrument characteristics would be tracked and corrected by mercury and halogen lamp tests. We have seen examples of such long-term stability at the South Pole where the instrument was operated without any additional calibrations for seven years.

The requirement about frequent calibrations is mostly based on the need of regular instrument maintenance that many operators cannot carry out by themselves (and to characterize the changes of the slits). In the case of the triad, such maintenance is done regularly. Between the calibrations, the constants were tracked by the lamp tests and the ETC was adjusted accordingly.

When the primary calibration has been done for one of the reference instruments at MLO, this instrument can be used to validate the status of other reference instruments in Toronto. So, to satisfy the 2-3 years interval between the calibration requirement, it is sufficient if at least one triad Brewer is calibrated at MLO every 2-3 years.

3. The transfer method from the triad to the travelling reference need to be clarified. Which of the instruments are used for transfer? What ozone data do you use for the transfer? That from the BrT or the straylight-free data? The observations from the BrT, BrT-D or an average of all six instruments? Which period of time is used for the calibration of the traveling reference.

The calibration process for the traveling references is the same as for any transferred calibration (Savastiouk, 2006): the instrument to be calibrated is assessed to make sure that its hardware is working properly, all the necessary characterization tests are done, simultaneous direct-sun data are collected with the triad instruments, and an average of BrT is used using $1.2 \leq \mu \leq 3.2$ for TCO $\leq 350$ DU to establish the ETC. However, the ozone calibration transfer is beyond the scope of this paper.

4. Different updated versions of the model Fioletov et al. [2005] have been used to establish the performance of the Brewer instrument, but this method is not used for the satellite and reanalysis comparison. For validation of this model a comparison of the triads using hourly observations (as reanalysis ) may be of interest.

In this work, we used three statistic models. Models 1 and 2 used here are strictly following the model designs described in Fioletov et al. 2005. Model 3 is a new one, or more precisely a modified Model 2. Here, Model 1 was used to directly assess the performance of reference instruments by examining their measured TCO values. Models 2 and 3 were used to examine the errors of ETC and the effective ozone absorption coefficient. The major difference between Models 2 and 3 are that they are using different "baseline" ozone. The former one uses the baseline ozone values derived from Model 1; the latter one uses the values from Pandora measurements.

We thank the referee for the suggestion to compare triads with satellite and reanalysis data using Model 3. The referee is correct that for Model 3, we can use any baseline ozone, as long as it is from a third party. However, we should note that this baseline ozone should have equivalent or better sampling frequency than that from Brewers. Given that satellites have a much lower frequency (about daily), we did not use them in Model 3. It might be possible to use the hourly reanalysis data, but the model also has uncertainties propagated from its assimilation sources (e.g., see Fig. 7, the "shift" in 2004). Thus, using reanalysis data would make the assessment for triad more complicated (i.e., we cannot easily separate the errors from the reanalysis model, satellite instrument, and Brewers). On the other hand, Pandora was selected because of its good precision (about 0.5 DU, see Zhao et al. 2016) and high sampling frequency (less than 5 minutes).

5. The Methods 2 and 3 also evaluate the error in the Extraterrestrial constant and absorption coefficient. These parameters are also obtained during the calibration, but no comparison is made between the model-derived parameters and those obtained when the instrument is calibrated.

As provided in previous responses, Models 2 and 3 do not generate an estimation of ETC or effective ozone absorption coefficient themselves, but their estimated errors. Thus, the model-derived parameters were used to evaluate the performance of the Brewers but not directly compared with those calibration constants (ETC or effective ozone absorption coefficient). As discussed in Sections 4.1.2 and 4.1.3, by the nature of errors in ETC and or effective ozone absorption coefficient, they may compensate each other and produce "reasonable" final TCO data products, despite the errors of themselves might be relatively large. Thus, we recommend not only examining the deviations of TCO values (e.g., Model 1), but also performing suggested Models 2 and 3 analyses for Brewer triads. An example of this practice was provided in Appendix B.

6. The Stray light effect on the ozone is the power law of the ozone slant column Karppinen et al. [2015] Moeini et al. [2019], although the observations are limited by air mass (3.5) and not by ozone slant column. A Brt to BrtD comparison against the ozone slant column may give us the correct limitation of the ozone slant column for the analysis.

[Figure]

**Figure R3. The relative difference between BrT and BrT-D, in terms of air mass factor ($\mu$) and slant column ozone. The error bars represent $1\sigma$ of the relative difference values. The black dash lines show the -1 % relative difference.**

We fully agree with the referee that the slant column is a governing factor. The 3.5 limits of air mass factor were examined and validated by previous works conducted in Toronto with Brewers, which should address the stray light effect in single Brewers sufficiently. As suggested by the referee, analysis of the percentage difference between BrT and BrT-D is provided in Fig. R3, in term of a function of both air mass factor ($\mu$) and slant column ozone. In general, they provided the same picture, i.e., BrT and BrT-D start to have more than 1% relative difference when $\mu > 3.5$ (equivalent to slant ozone 1200 DU).

Unless stray light correction is implemented, a filter based on slant ozone amount is impractical as it can easily allow poor data to go through if TCO is high, e.g., TCO = 300 DU ($\mu$ = 4 and slant column = 1200 DU) will be calculated as TCO = 250 DU ($\mu$ = 4 and slant column = 1000 DU) if stray light is present and pass the filter of 1000 DU. Moreover, any filtration based on slant column may introduce a bias in the data since low values would pass through the filter, while high values would not.

The manuscript has been revised to include some of these information and the Figure R3 has also been included in the supplement file (as Fig. S2).

*It is found that the air mass dependencies of BrT and BrT-D are consistent within these two periods.*

*Further information on relative difference between BrT and BrT-D, in terms of air mass factor and slant column ozone are provided in Fig. S2.*

7. The use of different timescales, monthly, three monthly or six monthly make the comparison of the different models difficult Please unify the results.

Following this suggestion, analyses are now using a consistent 3-month frequency.

[Figure]

***Figure 3. 3-month** relative differences between Brewers and Pandora total column ozone. **3-month** averages are calculated if there are at least ten coincident measurements between Brewer and Pandora for that period. The black dash line represents the time when BrT-D was relocated to Egbert, i.e., Pandora and BrT-D were not co-located.*

[Figure]

*Figure 4. Relative systematic uncertainties in ETCs and effective ozone absorption coefficients estimated using Model 3. Description of y-axes is in Fig. 2. Each point on the graph represents a 3-month average. The black dash line represents the time when BrT-D was relocated to Egbert.*

8. Results of the regular standard lamp tests of the Brewers, normally a good indicator of the stability of the instrumental calibration. A comparison of these measured SL-test records with the presented statistical parameters should be included and hopefully show the same good stability.

The measured SL-test records are included in the data processing. Thus, the ETC values have been corrected based on the SL tests. The SL test is not a measure of data quality, but a measure of instrument's spectral sensitivity changes that are applied to the data processing. Having relatively stable SL results are of little importance if not properly used in data processing and, conversely, even large variability in SL test results can be successfully used to correct the data (Lam et al., 2007). As the SL corrections have been made within BPS and the Model 2 used BPS outputs as input, directly comparing the SL records and Model 2 outputs may not be very meaningful. We have included more discussions about the data quality assurance in the revised manuscript.

*The way that the data are processed also affects the results. Siani et al, (2018) concluded that the ozone data processed by different software agree at the 1 % level; however, some differences can be found depending on the software in use. They also recommended "a rigorous manual data inspection" of the processed data and to be careful with how Standard Lamp (SL) test results are used. Visual data screening was also used by Stübi et al., (2017b) to eliminate outliers. However, this approach raises the*

*question of reproducibility of the obtained results and must be carefully documented. For BrT and BrT-D's data reprocessing, we recommend using the statistical models developed in relevant studies to help the identifications of potential hardware or software issues. To keep the integrity of the world reference instruments, data reprocessing could be done only if solid evidence of imperfection of hardware or software been found and confirmed by Brewer technicians and researchers.*

Specific Comments

3.1. Page 1 Line 27 Reference to the WMO requirements document is missing.

The reference to WMO has been included.

*The random uncertainties of individual reference instruments are within the WMO/GAW requirement of 1 % (WMO, 2001; 0.49 % and 0.42 % for BrT and BrT-D, respectively as estimated in this study).*

3.2. Page 1 Line 27 Reference to the uncertainty analysis is missing.

The uncertainty analysis, i.e., 0.49% and 0.42% reported here, was made by this work.

*The random uncertainties of individual reference instruments are within the WMO/GAW requirement of 1 % (WMO, 2001; 0.49 % and 0.42 % for BrT and BrT-D, respectively as estimated in this study).*

3.3. Page 2 Line 49 random uncertainty? Please use standard meteorologic terminology

*Data analysis from this study shows that the precision of individual observations are within ±1 % in about 90 % of all measurements.*

3.4. Page 2 Line 53 Please update Stray Light correction references, [Karppinen et al., 2015], [Rimmer et al., 2018]

New references have been included.

*Internal instrumental stray light affects measurements made with the single-monochromator instruments; therefore, corrections are applied to the data when necessary (Bais et al., 1996; Fioletov et al., 2000; Karppinen et al., 2015; Rimmer et al., 2018).*

3.5. Page 3 Line 63 The Arosa triad is now in Davos at PMOD World Radiation Center([Stübi et al., 2017])

This information has been updated.

*The Arosa triad (Staehelin et al., 1998; Stübi et al., 2017b), formed in 1998, was the second Brewer triad worldwide (composed of two Mark II and one Mark III instruments; now in Davos at PMOD World Radiation Center (Stübi et al., 2017a)).*

3.6. Page 3 Line 65 Reference comparisons are described in [Redondas et al., 2018]

This information has been updated.

*The regional reference instruments are regularly compared to the world reference instruments via a travelling standard (Redondas et al., 2018).*

3.7. Page 3, Line 80 The instrument calibration every 2-6 years ?, the range looks 3-8 years.

 This information has been updated.

*Each individual reference instrument is independently calibrated at the Mauna Loa Observatory (MLO), Hawaii (19.5° N, 155.6° W, 3400 m asl), every 3-8 years (see Table 1) via the Langley plot calibration method.*

3.8. Page 5, Line 116 Please detail the configuration of the BPS. Was this software also used for the previous Fioletov et al. [2005] analysis? Which are the main differences?

The software used in this study is the same as that used in Fioletov et al. (2005). The text has been revised to reflect this information.

*Brewer data was processed by Brewer Processing Software (BPS) developed by ECCC (Fioletov and Ogyu, 2008). The same processing software was used in Fioletov et al. (2005).*

3.9. Page 5, Line 113 Please associate the references with the corresponding product

The sentence has been modified to associate the references with corresponding products.

*The Brewer spectrophotometer provides data products that include column ozone (e.g., Kerr, 2002; Kerr et al., 1981), column sulphur dioxide ($SO_2$; e.g., Fioletov et al., 1998; Zerefos et al., 2017), column nitrogen dioxide ($NO_2$, by Mark IV only; e.g., Cede et al., 2006; Kerr et al., 1988), spectral UV radiation*

*(e.g., Bais et al., 1996; Fioletov et al., 2002), aerosol optical depth (e.g., Kazadzis et al., 2005; Marenco et al., 2002), and effective ozone layer temperature via group-scan technique (Kerr, 2002).*

3.10. Page 6, Line 140 What is an independent calibration technique? Please clarify.

Additional information has been included.

*As previously described, to maintain the high precision of all Brewer instruments (i.e., transfer the $F_0$ value), the world reference instruments (BrT and BrT-D) receive their $F_0$ values via the independent calibration technique. In short, these high-precision $F_0$ values were determined by fitting the measured F values as a linear function of air mass factor (see Eqn. 1). For example, in clear sky conditions with stable ozone values, if measurements are made under a range of air mass factors throughout a day, then the intercept of the linear fitting of $(F + \Delta\beta m)$ versus $\mu$ will be $F_0$. More technical details, such as calibration periods, averaging, and why MLO is the ideal site for this practice are provided in details in Kerr 2010.*

3.11. Page 7, Line 180 Please indicate the Pandora calibration.

Unlike Brewers, Pandora instruments do not need to perform the independent calibration at MLO. Some of these details, e.g., construction of extraterrestrial spectrum, were provided in the first paragraph of this section (Section 2.2). A new sentence has been included to indicate the Pandora calibration as suggested.

*The Pandora and BrT-D instruments have good stray-light control, and their air mass dependence is comparably low up to 81.6° SZA (within 1% up to AMF = 5.5; Zhao et al., 2016). Benefitting from the TOAS technique, unlike Brewers, Pandora instruments do not need the independent calibration at MLO (Tzortziou et al., 2012).*

3.12. Page 8, Line 165 Are Serdyuchenko cross sections used in this work? Please clarify.

All Brewer data used in this study are based on Bass-Pour 1985 effective ozone absorption coefficient. Although Serdyuchenko cross sections are recommended, they have not been widely implemented on the global Brewer network yet. Pandora (entire PNG) data was using Serdyuchenko cross sections.

*Another major difference between the Brewer and Pandora retrieval algorithms is their selection of ozone cross-section, i.e., the Brewer uses BP (Bass-Paur) ozone cross-section (at 228.3° K, Bass and Paur, 1985) and the Pandora uses* Serdyuchenko *ozone cross-section (at 225° K, Serdyuchenko et al., 2014).*

3.13. Page 8 , Line 170 Can you please summarize the differences between the official Pandora observations at Downsview that can be obtained from the Pandonia Global Network, and the ones used in this work? Are the observations used here also publicly available?

The effective temperature-corrected Pandora TCO data is not available on PGN. The major difference between the official Pandora TCO and the corrected TCO is their temperature sensitivity using empirical formula as described by Zhao et al., (2016). We have modified the sentence to summarize the differences between the official and corrected Pandora TCO. We have upload the corrected TCO data to ECCC's public data server and can be downloaded from:

https://collaboration.cmc.ec.gc.ca/cmc/arqi/Zhao_et_al_amt-2020-324/

*The effective temperature was calculated from temperature and ozone profiles provided by ERA-Interim (Dee et al., 2011).* In general, after correction, the multiplicative bias in Pandora ozone data can be decreased from 2.92 to -0.04 %, with the seasonal difference (estimated with monthly data) decreased from ±1.02 to ±0.25 % (see Fig. 11 in Zhao et al., 2016; i.e., comparing to Brewer, corrected Pandora data has -0.04 + 0.25% offset in summer and -0.04 – 0.25% offset in winter).

3.14. Page 8, Line 170 StrayLight (ozone slant column dependence), see general note 9.

The general note 9 is missing in the referee's report. Line 170 is not directly related to stray light. The information of Pandora stray light was discussed in comparison with Brewers in Appendix A.

3.15. Page 10, Line 220 Can you quantify the good quality of MERRA total ozone, for example, the BIAS and standard deviation with ground base?

The relative differences, biases, and standard deviations between Brewers and MERRA-2 were provided in Section 4.2 (see Fig. 7 and Table 6).

3.16. Page 10, Line 245 "the baseline is only needed to adjust for the time difference in ozone measurements by individual Brewers" How large is the time difference between the measurements of

the Brewers of the Triads? Couldn't they run in sync? If all the Brewers were in sync, would the baseline calculation (Ai coefficients still be needed?

It is correct that if all reference instruments' observations are synchronized, then we will not need to follow the design of Model 1 (i.e., use Ai coefficients to evaluate the deviations). Normally, Brewers can have one DS ozone observation about every 4 to 5 minutes, so theoretically it is possible. However, depending on the measurement schedules, Brewers may be operated in several different modes. For example, we plotted the DS TCO measurement intervals (i.e., the time gap between two successive DS TCO observations) in Fig. R7. It shows that the true DS TCO observation intervals can vary from about 5 to 30 minutes. Only less than 50 % of the observations were made with a "perfect" time interval, i.e., about 5 min. Also, a complete synchronization of schedules is not possible since the instruments perform different tasks, e.g., Brewer #015 is used for Umkher measurements and Brewer #014 is the main instrument for spectral UV measurements at Toronto.

[Figure]

**Figure R7. Probability of Brewer reference instruments' direct-sun TCO observation interval.**

3.17. Page 12, Line 307 As the triads receive its ETC independently, can be used as ozone for the model 3.

The referee is correct that a Brewer that received ETC via independent calibration can be used in Model 3 to provide the "baseline". However, we are reluctant to do this due to the sampling issue discussed in the previous question. Also, as the goal of this work is to assess the performance of all six reference instruments, we think that using one of them as a "baseline" is not ideal.

3.18. Page 13,Line 325 The total ozone above 400 DU are usual in Toronto and with 3.5 airmass limit means 1400 ozone slant column, so this observations are seriously affected by stray light. Why the double brewer are also limited in airmass?

A 3-month time series and histogram of TCO and air mass factor in Toronto were made when answering another question from referee #1. Here we presented it again below (see Fig. R3). For Toronto, the median TCO values are about 330 DU. Also, the stray light effect has been discussed and proved to be low with current selection of filters (see previous answers and Appendix A). We should point out again that unless stray light correction is implemented, a filter based on slant ozone amount is impractical as it can easily allow poor data to go through if TCO is high.

The referee is correct that double Brewers have much better stray light control and can provide data up to air mass factor 5 (see Figs. A2 and R3). However, in this work, since we want to provide the same assessment for both BrT and BrT-D, the same filtrations were made, i.e., air mass factor ≤ 3.5. However, the stray light performance of BrT and BrT-D was also examined and discussed in Appendix A.

[Figure]

Figure R6. Time series and histogram of TCO and air mass factor (μ) in Toronto.

3.19. Page 14, Line 341 σ' is not defined is it the mean? In that case, it would be better to use $\bar{\sigma}$

The symbols have been updated.

*The standard deviations (σ) of the 3-month averages plotted in Fig. 1a are 0.43 %, 0.36 %, and 0.42 % ($\bar{\sigma} = 0.40\%$) for Brewers #008, #014, and #015, which are comparable to the reported values from 1984 to 2004 (0.40 %, 0.46%, and 0.39 %). The double triad also shows good long-term stability with the Model 1 analysis, where all measurements are within ±1% compared to its baseline. The standard deviations are 0.44 %, 0.26 %, and 0.33 % ($\bar{\sigma} = 0.34\%$) for Brewers. #145, #187, and #191. From this, assuming that the instrument uncertainties are independent, the standard uncertainty of Brewers (δ) can be estimated as $\sqrt{1.5}\bar{\sigma}$, i.e., 0.49 % and 0.42 % for BrT and BrT-D, respectively.*

3.20. Page 15, Line 358 It looks like there is a factor 10 missing on the formula.

R6 is simply a linear combination of measured intensities in a modified scale. Depending on the values' scale, it may or may not need to be divided by a factor of 10 to get column ozone value. In the Brewer software, R6 is a linear combination of 10000×log(I) so to get ozone in DU it need to divide by 10. To make this more clear and consistent with Eqn. 1, the formula has been modified. More description of the models has also been included.

*Here, the errors in the ETCs and effective ozone absorption coefficients are estimated in R6 ratio units (the units used in the actual Brewer processing algorithm; R6 values corresponding to measured slant column, i.e., $\Omega = \frac{(R6 - ETCO_3)}{10\Delta\alpha\mu}$ in DU; ETCO₃ = -10⁴×F₀). The errors are converted from R6 ratio units to percentages of total column ozone by using typical conditions for Brewer measurements in Toronto (i.e., Ω = 330 DU, $\Delta\alpha$ = 0.34, and μ = 2), to provide more straightforward values to assess the impact of errors in the ETCs and effective ozone absorption coefficients. For example, if we have a model estimated error of ETCO₃ as 50 R6 ratio unit, it will correspond to $\frac{X}{10\Delta\alpha\mu\Omega} = 2.2$ % of total column ozone using the typical conditions described above.*

3.21. Page 15, Line 360 For the uncertainties of ETC, the goal is to have it within ±5 R6 ratio units. Please can you clarify which are the typical conditions, and how are these threshold parameters are obtained?

The goal in a calibration is to have an uncertainty in ETC of less than 1% effect on the TCO. Having an average TCO = 300 DU, $\Delta\alpha$ = 0.33 and $\mu$ = 1 (worst case for error), we can calculate that a 5 unit uncertainty in ETCO$_3$ gives about 1.5 DU, or 0.5% uncertainty in ozone.

**3.22. Page 15, 370 The goal of ETC and ozone absorption coefficient should be plotted also as reference.**

The real output of Brewer is TCO, and the goal of its random error is ±1 %. This is achieved by high precision ETC and $\Delta\alpha$ together. The estimates of ETC and $\Delta\alpha$ errors just provide values that best distribute the fitting residuals between "baseline ozone" (from Model 1) and measured ozone from one instrument. This means that the estimated errors here are the upper limits of the real errors within the ETC and $\Delta\alpha$. As shown by Fig. 2, these two estimated errors will compensate each other and make the "combined" error of TCO (or more precisely, the real error of TCO) within ±1 %. Thus, plotting the goal of ETC and $\Delta\alpha$ on Fig. 2 will be misleading, e.g., reader might think the calibration results failed to meet the goal.

**3.23. Page 16, Figure 2 Could you add the calibration dates to this figure For Brewer #008, it looks like the error is increasing over the last three years of the period between the 2008 and 2015 calibrations?**

Figure R8 has been made below with primary calibration dates for Brewer #008 included. Though the Triad instruments are exceptionally important and we attempt to ensure they run without a flaw, we cannot prevent component failure. In this case, the analog-to-digital (A/D) board failed just before the dates indicated by the referee (see more details in the supplementary information). All indications were that the replacement of the A/D board returned the instrument to normal operation and the replacement of this particular board does not affect Brewer characteristics; however, the instrument was disturbed. It was opened for repair, which does open the possibility for some issues, such as humidity changes (which may initiate a change in the NiSO4 band pass filter). These conditions would not be easily noted, especially when the differences between reference instruments were within ±1 %. The 2015 instrument review is a normal review of the instrument in preparation for absolute calibration. Doing the review before-hand minimizes instrument refurbishment time and maximized absolute calibration measurements in MLO.

[Figure]

**Figure R8. Modified Fig. 2, with primary calibration dates for Brewer #008 indicated on panel (a) by vertical dash lines.**

We have adjusted the figure to have larger fonts. Following the suggestion from referees, the figure also was updated with the new analysis frequency, i.e., 3-month.

[Figure]

*Figure 3. **3-month** relative differences between Brewers and Pandora total column ozone. **3-month** averages are calculated if there are at least ten coincident measurements between Brewer and Pandora for that period. The black dash line represents the time when BrT-D was relocated to Egbert, i.e., Pandora and BrT-D were not co-located.*

**3.25. Page 18, Line 420 Table 4: for comparison, we suggest to include the results of Model 2**

Following the suggestion, the results of Model 2 are also included in Table 4. Please note here Model 3 results have also been updated since Fig. 4 has been changed to have a 3-month analysis frequency.

*Table 4a. Mean errors of Δα and ETC for Brewer reference instruments (2013-2019) estimated with Model 3.*

| Brewer serial no. | Mean error of Δα [R6 absorption unit] | Mean error of ETC [R6 ETC unit] | Mean error of Δα* [%] | Mean ETC-related error# [%] |
|---|---|---|---|---|
| #008 | -0.0002 | -1.77 | -0.07 | -0.08 |
| #014 | 0.0051 | -32.87 | 1.50 | -1.45 |
| #015 | -0.0001 | -15.64 | -0.03 | -0.69 |
| #145 | 0.0007 | -8.01 | 0.21 | -0.35 |
| #187 | 0.0043 | -26.84 | 1.27 | -1.19 |
| #191 | 0.0039 | -23.27 | 1.15 | -1.03 |

*\* Mean % error in total column ozone, related to error in ozone absorptions; # Mean % error in total column ozone, related to error in ETC, corresponding to X when μ = 2, Δα = 0.34, and Ω = 330 DU (see Eqn. 3).*

*Table 4b. Mean errors of Δα and ETC for Brewer reference instruments estimated with Model 2.*

| Brewer serial no. [period] | Mean error of Δα [R6 absorption unit] | Mean error of ETC [R6 ETC unit] | Mean error of Δα* [%] | Mean ETC-related error# [%] |
|---|---|---|---|---|
| #008 [1999-2019] | -0.0011 | 6.79 | -0.33 | 0.30 |
| #014 [1999-2019] | -0.0005 | 3.26 | -0.15 | 0.14 |
| #015 [1999-2019] | 0.0006 | -3.79 | 0.17 | -0.17 |
| #145 [2013-2019] | -0.0011 | 5.68 | -0.33 | 0.25 |
| #187 [2013-2019] | 0.0026 | -0.61 | 0.08 | -0.03 |
| #191 [2013-2019] | 0.0026 | -1.05 | 0.08 | -0.05 |

**3.26. Page 19, Figure 5 It is difficult to see anything, probably it would be better to have one plot for every satellite.**

Following the suggestion, we made the plot that grouped time series by satellites (see Fig. R9). However, given the number of satellites and ground-based instruments included in this work, it is still very difficult to distinguish the difference between each pair. Also, the focus of this study is assessing the performance of Brewer reference instruments, thus we still prefer the original figure which is ground-based instrument oriented. We think Fig. R9 is also very useful when assessing the performance of each satellite data products. However, we think this is beyond the scope of current study.

The more detailed analysis results were provided in Fig. 6 and Table 5. In addition, the purpose of Fig. 5 is to provide the time series for each instruments and to give a general indication to reader about the Brewers' performance.

[Figure]

**Figure R9. The relative difference between satellites and the world Brewer reference triads (BrT and BrT-D). Each point represents a 3-month average. Brewers and satellite data are paired with the criteria shown in Table 3.**

3.27. Page 20, Figure 6 Could you please add a plot with the standard deviation?

The standard deviations were reported in Table 5.

3.28. Page 23, Line 515 There is a correction method to account for the filter non linearity Rimmer et al. [2018] why is not applied?

Currently, the BPS does not have ETC corrections for different filters. The ETC values were generated via a modified Langley method, which fit measurements from various ND filters together. The fitting for the Langley can be done per filter, if data is available. However, this method has other issues such as sampling differences (i.e., less measurements with high ND filters in low SZA conditions). On the other hand, using results from the so-called FI test to correct ND filter non-linearity has been done since 1993 and was described in Savastiouk (2006), which is same as the correction method in Rimmer et al. (2018). As described in Savastiouk (2006), considerable tests were made (i.e., involving twenty instruments) and found the errors caused by this filter non-linearity is within ±20 DU. Although this number might sounds high enough, the ETC value (which has contributions from many ND filters) will have an overall partial compensation for the effect. In other words, if we apply the FI correction for well-maintained Brewers, we would only expect to see changes in the final data for about ±1.0 DU, or within ±0.4 %, for most extreme cases. Besides being responsible for the world reference instruments, the ECCC group maintains the largest number of Brewers within this community (i.e. more than 40 Brewers). Meeting the WMO/GAW requirement and performing centralized data processing with minimal intervention are critical to the ECCC Brewer programme. We fully agree with the referee that further data improvements, such as this ND filter non linearity correction would further improve the quality of the data.

In summary, the goal of this study is to evaluate the overall performance of the current long historical triad record, and any further data improvement can be performed when a higher-precision reprocessing is needed and called upon by WMO/GAW.

3.29. Page 24, Line 523 Please clarify, the instrument is not described before and it is not clear how affect to the ozone measurement and when this issue affects to the results.

The sentence has been modified as suggested.

*If the combined focus of the monochromator mirrors of the instrument (see Savastiouk, 2006 for more details of instrument's optical elements) is not optimized and the illuminated filament of the mercury bulb is located in a significantly different location than the illuminated filament from the original bulb, as*

*much as a 5 micrometer step (one micrometer step is 0.7 pm) change may be seen. For reference, the effective ozone absorption changes by approximately 1% every 3 steps, so a 5 steps shift, which is extreme, can give an error of almost 2% in TCO.*

3.30. Page 24, Line 530 See General comment 8

We have answered referee's general comment 8.

3.31. Page 24, Line 545 Please explain how Figure 8b is obtained.

The sentence has been modified to explain this.

*As shown by the green and purple lines, the Downsview Brewers were sampling stratospheric ozone over Hamilton, while the Egbert Brewers were sampling stratospheric ozone over west of Brampton (the Brewers' sampling areas were estimated with viewing geometry of Brewers and MERRA-2 ozone profiles, ground projections of the intersections between the Brewer's line-of-sight and the modelled stratospheric ozone layer; Brampton is about 30 km west of Downsview, Hamilton is about 70 km south-west of Downsview).*

3.32. Page 24, Line 575 The determination with the model 2 of the ETC and ozone absorption coefficient cannot be define as error budget- The results of the Model 2 are quite far from the goal (the axis limits of Figure 2 are +/- 100 R6 ETC units but the goal is +/-5 R6 units).

The goal of Model 2 is to identify a potential source of errors in total ozone (i.e., errors in ETC or ozone absorption coefficient). As for the goal in ETC values, 5 R6 units of ETC error corresponds to 0.25 % shift in ozone for $\mu = 2$ and ozone = 300 DU. Most of the data in Figure 2 are within ±20 units of ±1 % under the same conditions.

*Further detailed error  analysis shows the impacts of ETC and ozone absorption coefficients errors for both reference triads are within ±2 % when the statistical Model 2 is used.*

3.33. Page 26, Line 585 The uncertainty of the Brewer triad is not established on this work, only its long term precision. The highly precise "group scan" is not discussed on this work and shouldn't be in the conclusions.

The sentences have been modified as suggested by the referee.

The *precision* of the Brewer triad instruments are under 0.5 %, while the differences with the best satellite instruments and reanalysis data are close to or slightly lower than 1 %. Further improvement of Brewer total ozone observation precision may be limited by the present Brewer 5-wavelength algorithm and Brewer hardware itself.

---

## Author Comment (AC3) · 27 Dec 2020

**Response to SC1:**

We thank Dr. León-Luis et al. for their very helpful comments and fast response to the issue that we identified and reported in our paper. The results provided in the document are very important and we have adapted some of this information into the revised paper. Our responses are given below in black with the comments from León-Luis et al. in blue. The new/revised text in the modified manuscript is given in red (italicized).

**1 Comparison with the RBBC-E triad in León-Luis *et al.* (2018)**

In the paper the authors claim that the comparison with the RBCC-E Triad presented in León-Luis *et al.* (2018) should not be carried out because the calculation is not consistent with the results of Model 1 in the present paper by Zhao et al. Note Model 1 was proposed in Fioletov *et al.* (2015).

5 In León-Luis *et al.* (2018), we calculate a quadratic polynomial fit for every Brewer as

$$O_3 = A \text{ -i- } B \cdot (t - t_0) \text{ -i- } C \cdot (t - t_0)^2 \tag{1}$$

obtaining for each instrument the corresponding values of *A*, *B* and *C*. Model 1 in Fioletov *et al.* (2015) however calculates common *B* and *C* values for all instruments.

We take the opportunity of the open discussion of this paper to update the calculations of the RBCC-E Triad to be consistent

10 with Fioletov *et al.* (2015), and also to compare the results of both Eq. 1 and Model 1 from Fioletov *et al.* (2015).

Table 1 contains the 3-month standard deviation of the $A_i$ coefficients obtained when the RBCC-E data are re-evaluated using Model 1, together with our previous published results. As can be observed, the values for each Brewer change slightly, depending on the method applied. However, the mean value of the Triad is similar, 0.23% versus 0.27%. This result confirms that there is very little difference between both methods when are applied to the RBCC-E Triad data.

15 This point can be better understood with an example. Fig. 1 demonstrates the total ozone column recorded on November 16th, 2016 (Fig. 4 in León-Luis *et al.* (2018)), where the data have been fitted used the two methods previously described. Table 2 contains the *A*, *B* and *C* coefficients calculated by both methods. As can be seen, regardless of the method used,

**Table 1.** RBCC-E and World Reference Triads: 3-month standard deviation. We include the values of the World Reference Triad from Zhao *et al.* (2020) for comparison.

| | RBCC-E | | | World Reference | | |
|---|---|---|---|---|---|---|
| Brewer | $\sigma_{3month}$, Eq.1 | $\sigma_{3month}$, Model 1 | Brewer | $\sigma_{3month}$, Model 1 | Brewer | $\sigma_{3month}$, Model 1 |
| #157 | 0.20 1 | 0.19 | #008 | 0.43 (0.40) | #145 | 0.44 |
| #183 | 0.31 | 0.26 | #014 | 0.36 (0.46) | #187 | 0.26 |
| #185 | 0.29 | 0.23 | #015 | 0.42 (0.39) | #191 | 0.33 |
| Mean | 0.27 | 0.23 | | 0.40 (0.42) | | 0.34 |

[Figure]

Note: The standard deviations of Brewers 157 and 185 were interchanged in Table 5 of reference León-Luis *et al.* (2018)

**Figure 1.** Ozone values measured on November 16th, 2016, marked with circles. Solid and dotted lines correspond to the $2^{nd}$ order polynomial fitted using Eq. 1 (RBCC-E method) and Model 1 from Fioletov *et al.* (2015) (World Reference Model method), the Time units are the minutes from solar noon. The *A* coefficients calculated with both methods are also shown.

the derived *A* coefficients are very similar. Therefore, the mean daily value of the RBCC-E Triad, the relative errors for each instrument, and the standard deviation, calculated from these coefficients, should not differ significantly. Furthermore, the *B* and *C* coefficients calculated by both methods are similar, which suggests that the adjusted functions will exhibit the same behavior as shown the Fig. 1. In conclusion, although both calculation methods are not the same, the results in the case of the RBCC-E Triad are very close. A similar result is achieved when no mathematical adjustment is used and the mean from the simultaneous measurements is calculated directly.

**Table 2.** Coefficients calculated with the two methods for the RBCC-E Triad data of November 16th, 2016 A, B, and C coefficients

| | |
|---|---|
| RBCC-E | $A_{157} = 276.53$, $B_{157} = -0.0040$, $C_{157} = -4.855\text{e} - 5$
$A_{183} = 278.72$, $B_{183} = -0.0025$, $C_{183} = -6.337\text{e} - 5$
$A_{185} = 277.42$, $B_{185} = -0.0028$, $C_{185} = -6.033\text{e} - 5$ |
| World Reference | $A_{157} = 276.76$, $A_{183} = 278.60$, $A_{185} = 277.35$
$B = -0.0030$, $C = -5.8122\text{e} - 5$ |

**Table 3.** Percentage difference of the mean of the three instruments, mean and its standard deviation and the percentage of observations 1% 0.5% and 0.25% of the five minutes simultaneous measurements and daily mean

| | Brewer | Mean | σ | <1% | <0.5% | <0.25% |
|---|---|---|---|---|---|---|
| 5 min | #157 | -0.041 | 0.342 | 0.994 | 0.909 | 0.687 |
| | #183 | 0.023 | 0.372 | 0.991 | 0.900 | 0.701 |
| | #185 | 0.018 | 0.342 | 0.99 | 0.921 | 0.758 |
| daily | #157 | -0.002 | 0.245 | 0.999 | 0.979 | 0.816 |
| | #183 | -0.005 | 0.309 | 0.999 | 0.931 | 0.757 |
| | #185 | 0.007 | 0.267 | 0.992 | 0.954 | 0.866 |

Table 3 shows the mean and standard deviation of the ratios for the 5 minutes simultaneous measurements and daily mean 25 values and note that the standard deviations values in this table are fairly similar to those in Table 1, even though the periods used for the calculations are not the same (5 minutes, daily and 3 months).

Up to this point, we have shown that both methods produce a similar result. The difference between the standard deviation reported by both Brewer Triads could then be associated to others factors which have not been considered in these works, such as e.g. the intra-day ozone variability or the number of ozone (Direct Sun) measurements made per day at each station.

30    These factors can affect the robustness of the mathematical fits and, hence, introduce small differences between the calculated *A* coefficients that are difficult to evaluate for two stations so far apart.

We thanks the Izana Atmospheric Research Centre group from AEMET for providing such detailed recalculations for the RBCC-E data, with the accurate method proposed in Fioletov et al., 2005. We fully agree with the group that using the index matrix method or the simple second-order fitting would not alter the results of the estimated random errors for Brewer triads. As the most accurate and precise total ozone observation instruments, well-maintained Brewers can have less than 0.5% random uncertainties. The difference caused by the selected fitting

algorithm should be well within this uncertainty levels for most of the cases. We pointed this issue out due to two reasons, 1) a part of the results (in León-Luis et al., 2018) was referred to as the use of Model 1 designed in Fioletov et al., 2005, which is not accurate. We thank the AEMET group for providing these re-calculated comparable results. 2) Moreover, the design of Model 1 is just a part of the whole evaluation scheme that been proposed. Model 2 designed in Fioletov et al. 2005 needs a "baseline" ozone, which is calculated from Model 1. For cases illustrated by this comments/report (i.e., RBCC-E data in 2016 Nov. 16), the baseline for this day can be defined as $(A_{157} + A_{183} + A_{185})/3 + B(t-t_0) + C(t-t_0)^2$, if B and C term of the fitting is "common" factors shared by all three instruments. However, when one selects to use a simple second-order fit for each of the instrument, then one will have three B terms and three C terms. One may argue that we also can average B and C terms to receive a "baseline" ozone; however, for some case, the B and C terms can be very different from instruments to instruments (if one only apply the simple fitting). We want to emphasize that the design of Model 1 is only a starting part of the evaluation scheme. The designed models work together to provide a guide in evaluating the performance of Brewers, if no other high-quality reference data can be used as a "referee".

2 Additional comments

In this section we include some other comments to Zhao et al. (2020), but first we want to acknowledge the effort of the World Reference Triad to maintain all these instruments during decades with such a high precision. Once the precision of the Triads has been established the challenge is to quantify the uncertainty, especially that produced by the described instrumental issues and include them in the analysis.

We thank the AEMET group again for providing us with such important comments and suggestions. The collaborations within the Brewer network is important not just for Brewer researchers, but also for the global ozone monitoring activities and related ozone research studies. The ECCC group designed and maintained the Brewer instruments since the 1970s. Almost a half-century of dedications to ozone monitoring work is a big accomplishment made by all Brewer scientists, technicians, and managers, and more importantly from our collaborators. As noted in the table of world reference instruments' primary calibration trips, some of the reference instruments were calibrated at Izana with great help from the AEMET group.

1. We do not agree with the comment that the 0.5% level cannot be achieved due to limitations of the Brewer hardware. Some of the issues described, such as for example the filter non linearity, can be addressed, and indeed are accounted for

in the processing performed at Eubrewnet. Eubrewnet's processing also takes into account the issue described for Brewer #15 – the observations not compensated with mercury tests are automatically filtered out.

The purpose of this work is to evaluate the triad performance based on the existing calibration results. Numerous works have been done by researchers worldwide in past decades to improve Brewers' accuracy and precision. However, not all of them have been implemented to the current reference instruments' results. We also provided a more detailed reply on issues such as filter non-linearity correction to referee #2. ECCC group maintains the largest number of Brewers within this community (i.e. more than 40 Brewers). Meet the WMO/GAW requirement and performing centralized data processing with minimal intervention are critical to the ECCC Brewer programme. In summary, the goal of this study is to evaluate the overall performance of the current long historical triad record, and any further data improvement can be performed when a higher-precision reprocessing is needed and called upon by WMO/GAW. ECCC group welcomes and is looking forward to continue and further our collaborations with AEMET group on these activities in future.

2. The cited Pandora manual has more than 150 pages, so it is difficult to find the ozone processing details. It could be better to refer to the ozone processing in the user guidelines avaliable at https://www.pandonia-global-network.org/wp-content/uploads/2020/01/LuftBlick_FRM4AQ_PGNUserGuidelines_RP_2019009_v1.pdf. Furthermore, if we understand it correctly, the data used in the present paper by Zhao et al. is not the operational one that is available to the public for download.

We have included this in the reference. The effective temperature corrected Pandora ozone is not available on the PGN website. We have upload the corrected TCO data to ECCC's public data server and can be downloaded from: https://collaboration.cmc.ec.gc.ca/cmc/arqi/Zhao_et_al_amt-2020-324/

*Additional information on Pandora calibrations, operation, retrieval algorithms and correction method can be found in Cede (2019; Cede et al., 2019), Tzortziou et al., (2012), and Zhao et al., (2016).*

*Cede, A., Tiefengraber, M., Gebetsberger, M. and Kreuter, M.: TN on PGN products "correct use" guidelines, Pandonia Global Network. [online] Available from: https://www.pandonia-global-network.org/wp-content/uploads/2020/01/LuftBlick_FRM4AQ_PGNUserGuidelines_RP_2019009_v1.pdf (Accessed 13 November 2020), 2019.*

3. It looks that there is a trend on the Merra comparison from 2005 to 2015, with Brewer #015 going from +2% to -2%

We are not sure Brewer #015 is the only instrument that shows a "trend". For example, Brewers #008 and #014 look like they also have a decreasing trend in this period, just not as "strong" as the one observed by Brewer #015. It is difficult for us to determine if this trend is real. To continue this investigation, one will need to collaborate with the reanalysis group and investigate not just Brewers, but also any upgrade or modifications in the model and its source of assimilation. However, we think that this interesting topic could be a standalone research topic and is beyond the scope of the current study.

4. Appendix A. The standard deviation of the ozone measurement is strongly affected by clouds and is also used as cloud mask to filter the AOD measurements affected by rapid moving clouds (López-Solano et al., 2017). Some of the brewer are equipped with full sky cameras, are the observations reported in Zhao et al. (2020) also filtered by clouds?

The standard data screening procedure used the standard deviations of 5 individual Brewer measurements to remove DS values obtained under cloudy conditions or under moving clouds. In very rare occasions when the standard deviations of such measurements are within the established limits, but the Sun is obscured by the clouds, an additional screening is done using the absolute intensity at the longest wavelength (320nm). Sky cameras are not very useful for such screening since the sky cloud coverage does not really affects DS measurements unless the clouds block the Sun. Sky camera data was not used in this study.

5. Appendix A. Figure A2 shows the dependence with the ozone air mass factor (AMF), as the stray-light is a function of AMF (Karppinen et al. (2015)) , but in the text the discussion is focused on the solar zenith angle and air mass

Further discussions of the stray light issue were provided in our answers to referees #1 and #2. Please refer to our answers in those replies.

6. Appendix A. An statistical approach to estimate the single triad stray light Diemoz et al. (2015) or the determination of the empirical correction by comparison with the double one (Redondas et al., 2018) could be performed to the dataset.

We thank the AEMET group for this very useful suggestion. When re-processing of the Brewer reference instruments records are needed, we will try to implement these proposed methods.

---

## Author Response (AR2)

**Response to Referee #2:**

We thank referee #2 for their helpful comments. Our responses are given below in black with the referee's comments in blue. The new/revised text in the modified manuscript is given in red (italicized).

The time series of the observations as the source of the analysis must be on the main paper rather than on the supplement.

Done.

*The assessment of the Brewer reference instruments was performed using Models 1, 2, and 3 defined in Section 3. The time series of Brewer reference triads' total column ozone (TCO) observations in Toronto is shown in Fig. 1. To ensure the assessment is based on good quality data, the data were strictly filtered (i.e., data from single and double spectrometer instruments with reported standard deviation > 3 DU or μ > 3.5 are removed).*

[Figure]

*Figure 1. Time series of Brewer reference triads' total column observations observations in Toronto. Vertical black dash lines indicate the time of primary calibrations as shown in Table 2.*

The figures are difficult to view, lines connecting points should not be drawn when are missing observations in between like Figure 3. To improve the visibility I suggest extending the x-axes using the page width, increase the size of the symbols and use different symbols for BrT and BrT-D.

Done.

There are several references to Savastiouk's 2006 thesis, it will be helpful for the reader if the chapter and/or pages are specified.

Done.

*For example, the effective ozone absorption coefficients (Δ$\alpha$) are determined for each individual instrument in laboratories via dispersion test, and are regularly checked using the stable solar spectrum as the reference using the so-called Sun Scan test (Savastiouk, 2006, Sect. 4.2).*

*Table 1. Specific features of single and double Brewer reference triads*

| | Single Brewer | Double Brewer |
|---|---|---|
| Model Version | Mark II | Mark III |
| Serial No.(s) | #008, #014 and #015 | #145, #187 and #191 |
| Start of triad observations | September 1984 | October 2013 |
| Optical and spectral characteristics | Single monochromator: a dispersing monochromator with an 1,800 line/mm holographic diffraction grating. | Double monochromator: a top dispersing monochromator with a 3,600 line/mm holographic grating, and a bottom recombining monochromator that is a mirror image of the dispersing monochromator |
| | Spectra measured by a single monochromator that is affected by the internal instrumental stray light in the UV region (Bais et al., 1996; Fioletov et al., 2000). | Significantly less instrumental stray light (out-of-band, stray light fraction $10^{-7}$) than in the single monochromators ($10^{-5}$) (Fioletov et al., 2000). Thus, increased accuracy of ozone and UV measurements under certain conditions (Bais et al., 1996; Wardle et al., 1996). |
| Output | Solar radiation at six UV wavelengths is measured with the spectrometer. The wavelengths are 303.2 nm (almost exclusively for wavelength calibration, i.e., spectral reference test) and five operating wavelengths (306.3 nm, 310.1 nm, 313.5 nm, 316.8 nm and 320.1 nm) used to measure total column ozone and sulphur dioxide using the sun, sky or near full moon as a light source. | |
| | Provides high-quality ozone measurements with a slant ozone column amount up to 1000 DU, which for the global average total ozone column of 300 DU corresponds to an ozone air mass factor of 3.33 and a solar zenith angle (SZA) of about 73° (Zanjani et al., 2019). | Provides high-quality ozone measurements with a slant ozone column amount up to 2000 DU, which for the global average total ozone column of 300 DU corresponds to an ozone air mass factor of 6.67 and a SZA of about 81° (Savastiouk, 2006, Sect. 4.4). |

*The Brewer retrieval algorithm removes effects that are linear as a function of the wavelength, but this offset may not be enough in some cases and a shift of up to a few DU in the retrieved ozone values can*

*occur as a result of a ND filter switch (e.g., from ND filter #1 in the early morning to ND filter #4 in the noon; Savastiouk, 2006, Sect. 4.3). Instruments with ND filters from the same manufactured batch will demonstrate almost identical spectral behaviour. Thus, these instruments may have very similar characteristics, and therefore, demonstrate high precision; however, they all may be affected by the same or similar hardware-related systematic errors. There are other hardware-related factors that affect the accuracy and precision of Brewer measurements. For example, a simple replacement of the mercury bulb that is used to ensure the instrument stability could affect total ozone measurements, creating jumps in the data record. The bulb change has the potential to affect the CalStep (calibration step, the optimal micrometer position found in the "Sun Scan" test; Savastiouk, 2006, Sect. 4.4) of the instrument. If the combined focus of the monochromator mirrors of the instrument (see Savastiouk, 2006, Sect. 4.1 for more details of instrument's optical elements) is not optimized and the illuminated filament of the mercury bulb is located in a significantly different location than the illuminated filament from the original bulb, as much as a 5 micrometer step (one micrometer step is 0.7 pm) change may be seen.*

The SL record is a good indicator of the stability of the instrument calibration and a key parameter on the data processing. Showing the time series of the SL record and the SL correction will to address the stability of the instruments.

To address the suggestion, the SL correction values are provided below. The measured SL-test records are included in the data processing. Thus, the ETC values have been corrected based on the SL tests. However, the SL test is not a measure of data quality, but a measure of the instrument's spectral sensitivity changes that are applied to the data processing. The SL test results do not always reflect the instrument's stability. Changes and repairs of the Brewer hardware also affect SL readings and therefore can be misleading. For this reason, the SL test results shown in Figure R1 are not included in the revised manuscript.

[Figure]

**Figure R1. Time series of Brewer SL corrections in R6 unit in Toronto. Each point on the graph represents a 3-month average.**